

**Temporal variability and driving factors of the carbonate system in the Aransas**
**Ship Channel, TX: A time-series study**
**Melissa R. McCutcheon[1], Hongming Yao[1,#], Cory J. Staryk[1], Xinping Hu[1]**
[1] Harte Research Institute for Gulf of Mexico Studies, Texas A&M University – Corpus
Christi, TX 78412, USA
[#] current address: Shenzhen Engineering Laboratory of Ocean Environmental Big Data
Analysis and Application, Shenzhen Institute of Advanced Technology, Chinese
Academy of Sciences, Shenzhen 518055, China
___________________________
*Correspondence to:* Melissa R. McCutcheon (melissa.mccutcheon@tamucc.edu)
**Keywords**: $p$CO$_2$, acidification, diel variability, seasonal variability, autonomous sensors





**Abstract**
Estuaries are complex systems with substantial heterogeneity in water chemistry,
including carbonate chemistry parameters such as pH and partial pressure of $CO_2$ ($pCO_2$),
because of the diversity of co-occurring biogeochemical processes. To better understand
estuarine acidification and air-sea $CO_2$ fluxes from estuaries, it is important to study
baseline variability and driving factors of carbonate chemistry. Using both discrete bottle
sample collection (2014-2020) and hourly sensor measurements (2016-2017), we
explored temporal variability, from diel to interannual scales, in the carbonate system
(specifically pH and $pCO_2$) at the Aransas Ship Channel located in northwestern Gulf of
Mexico. Using other co-located environmental sensors, we also explored the driving
factors of that variability. Both sampling methods demonstrated significant seasonal
variability at the location, with highest pH (lowest $pCO_2$) in the winter and lowest pH
(highest $pCO_2$) in the summer. Significant diel variability was also evident from sensor
data, but the time of day with elevated $pCO_2$/depressed pH was not consistent across the
entire monitoring period, sometimes reversing from what would be expected from a
biological signal. Though seasonal and diel fluctuations were smaller than many other
areas previously studied, carbonate chemistry parameters were among the most important
environmental parameters to distinguish between time of day and between seasons. It is
evident that temperature, biological activity, and tide level (despite the small tidal range)
are all important controls on the system, with different controls dominating at different
time scales. The results suggest that the controlling factors of the carbonate system may
not be exerted equally on both pH and $pCO_2$ on diel timescales, causing separation of
their diel or tidal relationships during certain seasons. Despite known temporal variability





on shorter timescales, discrete sampling was generally representative of the average
carbonate system and average air-sea $CO_2$ flux on a seasonal and annual basis based on
comparison with sensor data.

## 1. Introduction

Estuaries, the dynamic environments where the coast and freshwater inflows meet
the ocean, are economically and ecologically important because they are biological
hotspots, but they are also heavily influenced by anthropogenic activity. Because of the
diversity of co-occurring biogeochemical processes, estuaries experience substantial
spatial and temporal heterogeneity in water quality and chemistry, including carbonate
chemistry parameters such as pH and partial pressure of $CO_2$ ($pCO_2$) (Hofmann et al.,
2011; Waldbusser and Salisbury, 2014). Carbonate chemistry, or the speciation of
inorganic carbon in seawater, is important for two main reasons. First, $CO_2$ acidifies
seawater, whether it is a result of uptake from the atmosphere (generally acknowledged
as ocean acidification, or OA) or it is produced by biogeochemical processes in the water
(that may intensify or alleviate the effects of OA). This is problematic because
acidification can negatively affect marine organisms, especially those that construct
calcium carbonate shells and skeletons (Barton et al., 2015; Bednaršek et al., 2012;
Ekstrom et al., 2015; Gazeau et al., 2007; Gobler and Talmage, 2014). Second, the ocean
contributes substantially to the global carbon budget, which is important to understand
because of climate change implications. Despite the small surface area of coastal waters
relative to the global ocean, coastal waters are recognized as important contributors in
global carbon cycling (Borges, 2005; Cai, 2011; Laruelle et al., 2018).





While open ocean environments are relatively well studied and understood
regarding carbonate chemistry, acidification, and air-sea $CO_2$ fluxes, large uncertainties
remain in estuarine environments. Estuaries are challenging to fully understand because
of the heterogeneity between and within estuaries that is driven by diverse processes
operating on different time scales such as river discharge, nutrient and organic matter
loading, stratification, and coastal upwelling (Jiang et al., 2013; Mathis et al., 2012). The
traditional sampling method for carbonate system characterization involving discrete
water sample collection and laboratory analysis is known to lead to biases in average
$pCO_2$ and $CO_2$ flux calculations due to daytime sampling that neglects to capture diel
variability (Li et al., 2018). Mean diel ranges in pH can exceed 0.1 unit and single day
ranges can exceed 1 pH unit, with especially high diel variability in biologically
productive areas or areas with higher mean $pCO_2$ (Challener et al., 2015; Cyronak et al.,
2018; Schulz and Riebesell, 2013; Semesi et al., 2009; Yates et al., 2007). These diel
ranges can far surpass the magnitude of the changes in open ocean surface waters that
have occurred since the start of the industrial revolution and rival spatial variability in
productive systems, indicating their importance for a full understanding of the carbonate
system.
Despite the need for high-frequency measurements, sensor deployments have
been limited in estuarine environments (especially compared to their extensive use in the
open ocean) because of the challenges associated with varying conditions, biofouling,
and sensor drift (Sastri et al., 2019). Carbonate chemistry monitoring in the Gulf of
Mexico (GOM), and especially its estuaries, has been relatively minimal compared to the
United States east and west coasts. The GOM estuaries, where this study takes place,





currently have less exposure to concerning levels of acidification than other estuaries
because of their high temperatures (causing water to hold less $CO_2$ and support high
productivity year-round) and often suitable river chemistries (i.e. relatively high buffer
capacity) (McCutcheon et al., 2019; Yao et al., 2020). However, respiration-induced
acidification is present in both the open GOM (e. g., subsurface water influenced by the
Mississippi River Plume and outer shelf region near the Flower Garden Banks National
Marine Sanctuary) and GOM estuaries, and most estuaries in the northwestern GOM
have also experienced long-term acidification (Cai et al., 2011; Hu et al., 2018, 2015;
Kealoha et al., 2020; McCutcheon et al., 2019; Robbins and Lisle, 2018). This known
acidification as well as the relatively high $CO_2$ fluxes from the estuaries of the northwest
GOM (which may change our understanding of global estuarine contribution to the
carbon budget) illustrates the necessity to study the baseline variability and driving
factors of carbonate chemistry in the region. In this study, we explored temporal
variability in the carbonate system in Aransas Ship Channel—a tidal inlet in a semi-arid
region of the northwestern GOM—using both discrete bottle sample collection and
hourly sensor measurements, and we explored the driving factors of that variability using
data from other co-located environmental sensors.
**2. Materials and Methods**
*2.1 Location*

Autonomous sensor monitoring and discrete water sample collections for

laboratory analysis of carbonate system parameters were performed in the Aransas Ship
Channel (27˚50'17"N, 97˚3'1"W). The Aransas Ship Channel is one of the few permanent
tidal inlets that intersect a string of barrier islands and connect the GOM coastal waters



with the lagoonal estuaries in the northwest GOM (Fig. 1). The Aransas Ship Channel
provides the direct connection between the nwGOM and the Mission-Aransas Estuary
(Copano and Aransas Bays) to the north and Nueces Estuary (Nueces and Corpus Christi
Bays) to the south (Fig. 1). The tidal range in the region is small, with around 0.6 m tides
on the open coast and less than 0.3 m in the estuaries (Montagna et al., 2011). Mission-
Aransas Estuary (MAE) is fed by two small rivers, the Mission (1787 km$^2$ drainage
basin) and Aransas (640 km$^2$ drainage basin) Rivers (http://waterdata.usgs.gov/), which
both experience low base flows punctuated by periodic high flows during storm events.
MAE has an average residence time of one year (Solis and Powell, 1999), so there is a
substantial lag between time of rainfall and riverine delivery to the Aransas Ship Channel
in the lower estuary. A significant portion of riverine water flowing into Aransas Bay
originates from the larger rivers further northeast on the Texas coast via the Intracoastal
Waterway (i.e. Guadalupe River (26,625 km$^2$ drainage basin) feeds San Antonio Bay and
has a much shorter residence time of nearly 50 days) (Solis and Powell, 1999; USGS, 2001).

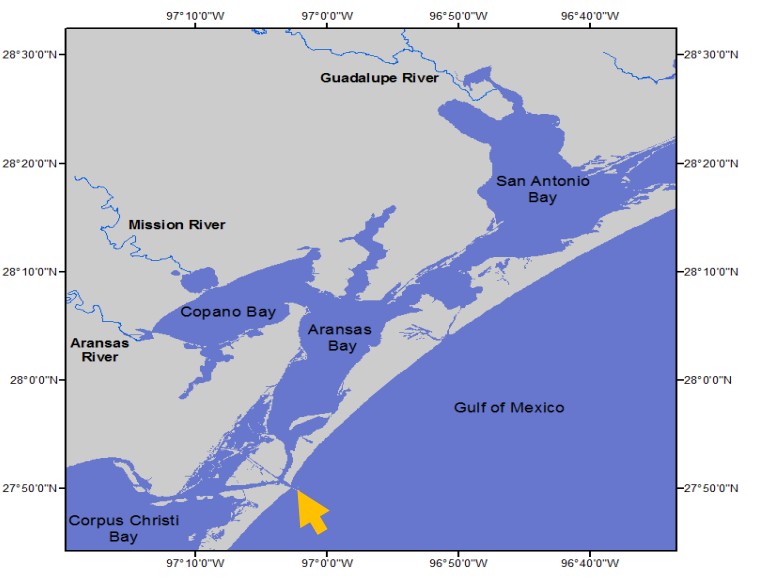

**Fig 1.** Location of Aransas Ship Channel where this study took place (arrow) and surrounding bay systems.

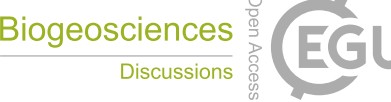

*2.2 Continuous Monitoring*


Autonomous sensor monitoring (referred to throughout as continuous monitoring)
of pH and $p$CO$_2$ was conducted from Nov. 8, 2016 to Aug. 23, 2017 at the University of
Texas Marine Science Institute's research pier in the Aransas Ship Channel. The sensor
deployment was shorter than intended because the pier where the sensors were deployed
was destroyed in the aftermath of Hurricane Harvey in 2017. The pH data were collected
using an SAtlantic$^®$ SeaFET pH sensor (on total pH scale) and $p$CO$_2$ data were collected
using a Sunburst$^®$ SAMI-CO$_2$. Temperature and salinity data were measured by a YSI$^®$
600OMS V2 sonde. Hourly data collected by all sensors (pH, $p$CO$_2$, salinity, and
temperature) were saved in the onboard data loggers and downloaded during service trips
to the field site. Sensor failures or pump failures occurred for short periods of time
throughout the deployment. Measurements were recorded on 262 individual days, with
176 of those days having the full set of 24 (hourly) measurements.
Ideally, *in-situ* sensors should be deployed under the sea surface. However, to
reduce the maintenance cost and effort for sensors deployed in warm water that
experiences intense biofouling, the sensors were set up to measure pH and $p$CO$_2$ from an
*ex situ* position using *in situ* seawater. Water was pumped from ~1 m below the sea
surface into the bottom spigot of a 100-Qt cooler that housed the SAMI-CO2 and SeaFET
sensors. To allow water outflow, a 1" hole was drilled at the opposite side of the spigot
near the top of the cooler rim, allowing water to flow back to the sea surface. The pump
was programmed to turn on 20 minutes before each hour, pumping more than enough
water to fully flush the cooler, and sensors recorded measurements on the hour. The YSI





sonde was deployed directly into the Aransas Ship Channel inside a 2" PVC pipe at ~1 m
below the sea surface.

Visits to the field site were conducted every two weeks to service all sensors and

clean the cooler. Additionally, duplicate water samples were collected on the hour during
service trips for quality assurance of sensor data and to check that surface water and
cooler chemistries aligned. Samples were collected from the channel near the pump inlet
and from the cooler that housed the sensors, and water temperature and salinity were
measured using a handheld YSI data sonde. The average difference between sensor pH
and laboratory pH (from the cooler) was used to establish a correction of -0.05 to the
final *in-situ* pH data since the SeaFET may experience drift. The difference between the
sensor $pCO_2$ and calculated $pCO_2$ is reported, but it is not used for a correction since the
spectrophotometric measurements of the SAMI-CO2 should not experience drift. Sensor
data were discarded from analysis during known periods of pump failure when cooler
chemistry separated from that of the Aransas Ship Channel.
*2.3 Discrete Sample Collection and Analysis*

In addition to the discrete sample collections that occurred for quality assurance

during sensor servicing visits, long-term monitoring via discrete water sample collection
was conducted at the Aransas Ship Channel from May 2, 2014 to February 25, 2020.
Sampling was conducted from a small vessel at a station very near to the sensor
deployment every two weeks during the summer months and monthly during the winter
months. Water sample collection followed standard protocol for ocean carbonate
chemistry studies (Dickson et al., 2007). Ground glass borosilicate bottles (250 mL) were
filled with surface water and preserved with 100 µL saturated mercury chloride ($HgCl_2$).





Apiezon ® grease was applied to the bottle stopper, which was then secured to the bottle
using a rubber band and a nylon hose clamp.
These samples were used for laboratory dissolved inorganic carbon (DIC) and pH
measurements. DIC was measured by injecting 0.5 mL of sample into 1 ml 10% $H_3PO_4$
(balanced by 0.5 M NaCl) with a high-precision Kloehn syringe pump. The $CO_2$ gas
produced through sample acidification was then stripped using high-purity nitrogen gas
and carried into a Li-Cor infrared gas detector. DIC analyses had a precision of 0.1%.
Certified Reference Material (CRM) was used to ensure the accuracy of the analysis
(Dickson et al. 2003). For samples with salinity>20, pH was measured using a
spectrophotometric method at $25 \pm 0.1°C$ (Carter et al. 2003) and the Douglas and Byrne
(2017) equation. Analytical precision of the spectrophotometric method for pH
measurement was ±0.0004 pH units. A calibrated Orion Ross glass pH electrode was
used to measure pH at $25 \pm 0.1°C$ for samples with salinity<20, and analytical precision
was ±0.01 pH units. All pH values obtained using the potentiometric method were
converted to total scale at *in situ* temperature (Millero 2001). Salinity of the discrete
samples was measured using a benchtop salinometer calibrated by MilliQ water and a
known salinity CRM.
*2.4 Data Processing and Statistical Analyses*
For the discrete samples, $pCO_2$ was calculated using CO2Sys for Excel.
Carbonate speciation calculations were done using Millero (2010) carbonic acid
dissociation constants ($K_1$ and $K_2$), Dickson (1990) bisulfate dissociation constant, and
Uppström (1974) borate concentration. Temporal variability was investigated in the form
of seasonal and diel variability (Tables 1-2). For seasonal analysis, December to February
was considered winter, March to May was considered spring, June to August was
considered summer, and September to November was considered fall. Two-way
ANOVAs were used to examine differences in parameter means between seasons, using
differences between monitoring methods as the second factor (as differences between
seasons may not be the same between monitoring methods, Table 3). Since there was a
significant interaction in the two-way ANOVA, the differences between seasons were
investigated within each monitoring method. Post-hoc multiple comparisons (between
seasons within sampling types) were conducted using the Westfall adjustment (Westfall,
1997). For diel comparisons, daytime and nighttime variables were defined as 09:00-
15:00 local standard time and 21:00-03:00 local standard time, respectively, based on the
6-hour periods with highest and lowest photosynthetically active radiation (PAR; data
obtained from the Mission-Aransas National Estuarine Research Reserve (MANERR) at
https://missionaransas.org/science/download-data. Paired $t$-tests, comparing the daytime
mean with the nighttime mean on respective days, were used to look for significant
differences between daytime and nighttime parameter values across the full sampling
period and within each season (Table 2).
Equation 1 was used for air-water $CO_2$ flux calculations (Wanninkhof, 1992;
Wanninkhof et al., 2009). Positive flux values indicate $CO_2$ emission from the water into
the atmosphere (the estuary acting as a source of $CO_2$), and negative flux values indicate
$CO_2$ uptake by the water (the estuary acting as a sink for $CO_2$).
$F = k\ K_0\ (pCO_{2,w} - pCO_{2,a})$ (1)





where k is the gas transfer velocity (in m $d^{-1}$), $K_0$ (in mol $l^{-1}$ $atm^{-1}$) is the solubility
constant of $CO_2$ (Weiss, 1974), and $p$$CO_{2,w}$ and $p$$CO_{2,a}$ are the partial pressure of $CO_2$ (in
µatm) in the water and air, respectively.

We used the wind speed parameterization for gas transfer velocity ($k$) from Jiang

et al. (2008) converted from cm $h^{-1}$ to m $d^{-1}$, which is thought to be the best estuarine
parameterization at this time (Crosswell et al., 2017) as it is a composite of $k$ over several
estuaries. The calculation of $k$ requires a windspeed at 10 m above the surface, so
windspeeds measured at 3 m above the surface were converted using the power law wind
profile (Hsu, 1994; Yao and Hu, 2017). To assess uncertainty, other parameterizations
with direct applications to estuaries in the literature were also used to calculate $CO_2$ flux
(Raymond and Cole 2001; Ho et al. 2006). We note that parameterization of $k$ based on
solely windspeed is flawed because several additional parameters can contribute to
turbulence including turbidity, bottom-driven turbulence, water-side thermal convection,
tidal currents, and fetch (Wanninkhof 1992, Abril et al., 2009, Ho et al., 2104, Andersson
et al., 2017), however it is currently the best option for this system given the limited
investigations of $CO_2$ flux and contributing factors in estuaries.

Hourly windspeed data used in calculations were retrieved from the NOAA-

controlled Texas Coastal Ocean Observation Network (TCOON;
https://tidesandcurrents.noaa.gov/tcoon.html). The closest station with windspeed data,
Port Aransas Station, was located directly in the Aransas Ship Channel (further inshore
than our monitoring location), however there were several long periods of missing data.
To fill in the data gaps, wind speed data from nearby Aransas Pass Station were used, and
for all subsequent gaps, data from nearby Nueces Bay Station were used. For continuous
monitoring data, TCOON's measured hourly windspeed at each time point was used in
flux calculations. For biweekly discrete samples, averaged daily windspeeds were
calculated from TCOON's measured hourly windspeeds and used in flux calculations for
the respective day. Monthly mean atmospheric $xCO_2$ data (later converted to $pCO_2$) were
obtained from NOAA's flask sampling network of the Global Monitoring Division of the
Earth System Research Laboratory at the Key Biscayne (FL, USA) station, when
available
(https://www.esrl.noaa.gov/gmd/dv/data/index.php?site=KEY¶meter_name=Carbon
%2BDioxide). For 2019 and 2020, when xCO2 data from Key Biscayne were
unavailable, monthly global average values were used
(ftp://aftp.cmdl.noaa.gov/products/trends/co2/co2_mm_mlo.txt).
*Factors controlling the carbonate system parameters*

Thermal versus non-thermal controls on $pCO_2$ were investigated following

Takahashi et al. (2002) over annual, seasonal, and daily time scales (Table 4). When
calculating annual T/B values with discrete data, only complete years (sampling from
January to December) were included. When calculating daily T/B values with continuous
data, only complete days (24 hourly measurements) were included.
$pCO_{2,thermal} = pCO_{2,mean} \times \exp[\delta \times (T_{obs} - T_{mean})]$                     (2)
$pCO_{2,nonthermal} = pCO_{2,obs} \times \exp[\delta \times (T_{mean} - T_{obs})]$                     (3)
Where the value for $\delta$ (0.0411 °C$^{-1}$), which represents average $[\partial \ln pCO_2 / \partial$
Temperature] from field observations, was taken directly from Yao and Hu (2017), $T_{obs}$ is
the observed temperature, and $T_{mean}$ is the mean temperature over the investigated time
period.



$$T/B = \frac{\max(pCO_{2,thermal}) - \min(pCO_{2,thermal})}{\max(pCO_{2,non-thermal}) - \min(pCO_{2,non-thermal})} \qquad (4)$$
Where a T/B greater than one indicates that temperature's control on $pCO_2$ is greater than
the control from non-thermal factors (i.e. physical and biological processes) and a T/B
less than one indicates that non-thermal factors' control on $pCO_2$ is greater than the
control from temperature.

Tidal control on parameters was investigated using only the continuous

monitoring data (Table 5). Hourly measurements of water level immediately offshore
from the Aransas Ship Channel were obtained from NOAA's Tides and Currents Aransas
Pass Station
https://tidesandcurrents.noaa.gov/waterlevels.html?id=8775241&name=Aransas,%20Ara
nsas%20Pass&state=TX. Tide data were merged with our sensor data by date and hour;
given that there were gaps in available water level measurements (and no measurements
prior to December 20, 2016), the usable dataset was reduced from 6088 observations to
5121 observations and fall was omitted from analyses. To examine differences between
parameters during high tide and low tide, we defined high tide as tide level greater than
the third quartile tide level value and low tide as a tide level less than the first quartile
tide level value. The difference between high and low tide for each parameter was
examined within each season (using t-tests) because of a significant interaction (based on
α=0.05) between the season and high/low tide factors in a two-way ANOVA.
**3. Results**
*3.1 Continuous monitoring results*

Over the 10-month continuous monitoring period, all sensor-measured parameters

showed substantial temporal variability on seasonal and diel time scales (Fig. 2, Tables 1-



3). Mean values of sensor-measured parameters over the entire monitoring period were:
temperature - 23.1°C ± 5.3°C, ranging from 9.4°C to 31.7°C; salinity - 30.8 ± 3.7,
ranging from 18.3 to 38.9; pH - 8.12 ± 0.10, ranging from 7.79 to 8.45; and $p$CO$_2$ - 416 ±
60 µatm, ranging from 251 µatm to 620 µatm (Table 1). Temperature was significantly
different between each season (Table 3), with the highest being summer and the lowest
being winter (Table 1). Salinity was highest in the summer and lowest in the fall, and
salinity differed between all seasons except from spring and winter (Tables 1 and 3). pH
and $p$CO$_2$ were both significantly different between all seasons (Table 3). Winter had
both the highest seasonal pH (8.19 ± 0.08) and lowest seasonal $p$CO$_2$ (365 ± 44 µatm)
and summer had both the lowest seasonal pH (8.05 ± 0.06) and highest seasonal $p$CO$_2$
(463 ± 48 µatm) (Table 1, Fig. 2- 3).
Table 1. Mean and standard deviation of annual and seasonal temperature, salinity, pH,
$p$CO$_2$, and CO$_2$ flux from continuous monitoring, discrete sampling over the continuous
monitoring period, and discrete sampling over the entire sampled period. Reported annual
means are seasonally weighted to account for disproportional sampling between seasons
(however, reported annual standard deviation is associated with the un-weighted,
arithmetic mean). CO$_2$ fluxes were calculated using the Jiang et al. (2008) wind speed
parameterization for gas transfer velocity, and ranges of CO$_2$ flux that are given in
brackets represent means calculated using parameterizations from Ho et al. (2006) and
Raymond and Cole (2001), respectively.

| Parameter | | Continuous Monitoring | Discrete Sampling | |
| --- | --- | --- | --- | --- |
| | Time Period | Nov. 8 2016 – Aug 23, 2017 | Nov. 8 2016 – Aug 23, 2017 | May 2, 2014- Feb. 25, 2020 |
| **Temperature** (°C) | **Annual** | 23.1 ± 5.3 | 23.5 ± 5.0 | 24.1 ± 5.3 |
| | Winter | 17.3 ± 2.1 | 17.3 ± 1.1 | 16.2 ± 2.0 |
| | Spring | 23.8 ± 2.8 | 23.4 ± 2.9 | 22.6 ± 3.7 |
| | Summer | 29.7 ± 0.8 | 29.6 ± 0.5 | 28.7 ± 1.4 |
| | Fall | 22.5 ± 2.1 | 23.6 ± 0.1 | 25.5 ± 4.5 |
| **Salinity** | **Annual** | 30.8 ± 3.7 | 30.4 ± 3.5 | 30.1 ± 4.4 |
| | Winter | 30.0 ± 3.7 | 29.3 ± 4.6 | 28.9 ± 2.9 |
| | Spring | 30.2 ± 2.6 | 30.0 ± 1.7 | 28.7 ± 3.4 |
| | Summer | 33.3 ± 3.2 | 33.6 ± 3.2 | 34.6 ± 2.8 |
| | Fall | 27.6 ± 3.7 | 28.8 ± 0.1 | 28.4 ± 4.5 |
| **pH** | **Annual** | 8.12 ± 0.10 | 8.092 ± 0.078 | 8.079 ± 0.092 |
| | Winter | 8.19 ± 0.08 | 8.157 ± 0.041 | 8.162 ± 0.065 |
| | Spring | 8.09 ± 0.09 | 8.078 ± 0.056 | 8.077 ± 0.066 |

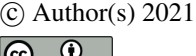



|  |  | | | |
| --- | --- | --- | --- | --- |
|  | Summer | $8.05 \pm 0.06$ | $7.999 \pm 0.051$ | $7.975 \pm 0.046$ |
|  | Fall | $8.18 \pm 0.05$ | $8.136 \pm 0.001$ | $8.100 \pm 0.071$ |
| $p$CO$_2$ (µatm) | **Annual** | $416 \pm 60$ | $400 \pm 71$ | $406 \pm 100$ |
|  | Winter | $365 \pm 44$ | $349 \pm 31$ | $331 \pm 39$ |
|  | Spring | $436 \pm 45$ | $413 \pm 54$ | $396 \pm 67$ |
|  | Summer | $463 \pm 48$ | $480 \pm 59$ | $511 \pm 108$ |
|  | Fall | $400 \pm 25$ | $357 \pm 2$ | $386 \pm 62$ |
| CO$_2$ Flux (mmol m$^{-2}$ d$^{-1}$) | **Annual** | $0.2 \pm 23.7$ $[0.1 - (-87.6)]$ | $-1.5 \pm 9.2$ $[(-2.6) - (-4.5)]$ | $(-0.8) \pm 18.7$ $[(-0.7) - 5.3]$ |
|  | Winter | $(-16.9) \pm 29.2$ $[(-14.6) - (-444.0)]$ | $(-9.9) \pm 5.2$ $[(-8.3) - (-16.2)]$ | $(-13.0) \pm 13.5$ $[(-10.6) - (-25.6)]$ |
|  | Spring | $7.6 \pm 15.0$ $[6.5 - 109.0]$ | $1.0 \pm 7.1$ $[1.0 - 3.3]$ | $(-6.5) \pm 12.2$ $[(-5.5) - (-18.0)]$ |
|  | Summer | $10.8 \pm 13.3$ $[9.1 - 28.9]$ | $10.5 \pm 7.8$ $[8.6 - 16.3]$ | $18.3 \pm 19.6$ $[15.3 - 65.5]$ |
|  | Fall | $(-0.9) \pm 7.7$ $[(-0.7) - (-44.0)]$ | $(-7.5)$ $[(-6.2) - (-11.4)]$ | $(-2.3) \pm 13.7$ $[(-1.9) - (-0.9)]$ |


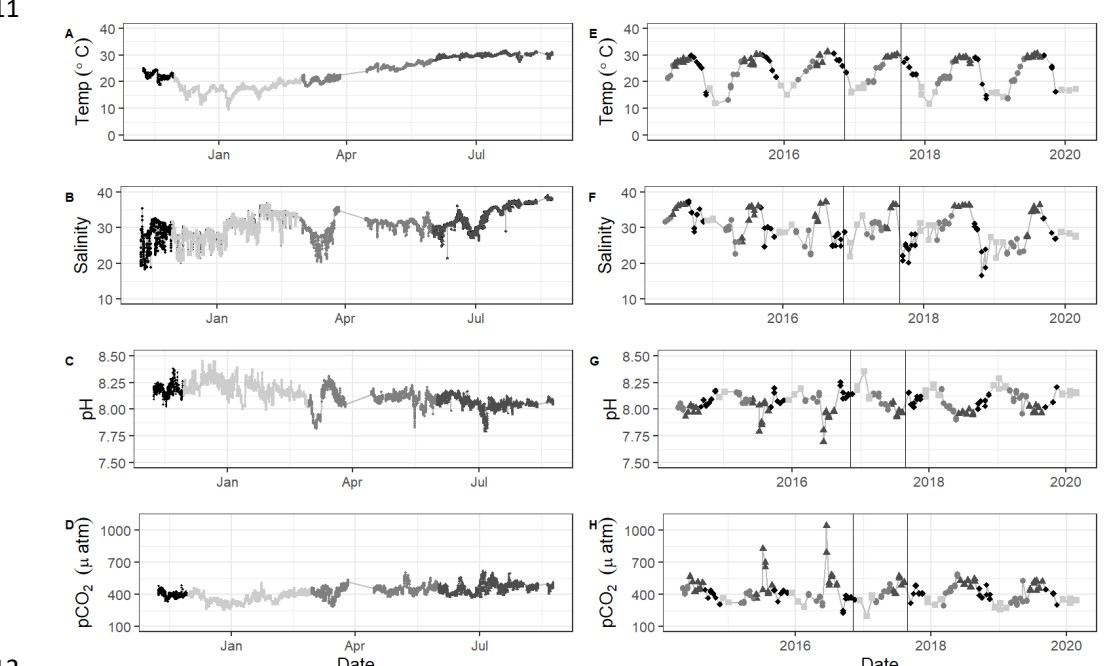

Fig 2. Time series data from continuous monitoring (A-D, Nov 8, 2016 to Aug 3, 2017)
and discrete sample analysis (E-H, Nov 8, 2016 to Aug 3, 2017) at the Aransas Ship
Channel. Gray scale (and shape) in the datapoints represents divisions between the four
seasons. Vertical lines in (E-H) denote the time period of continuous monitoring.





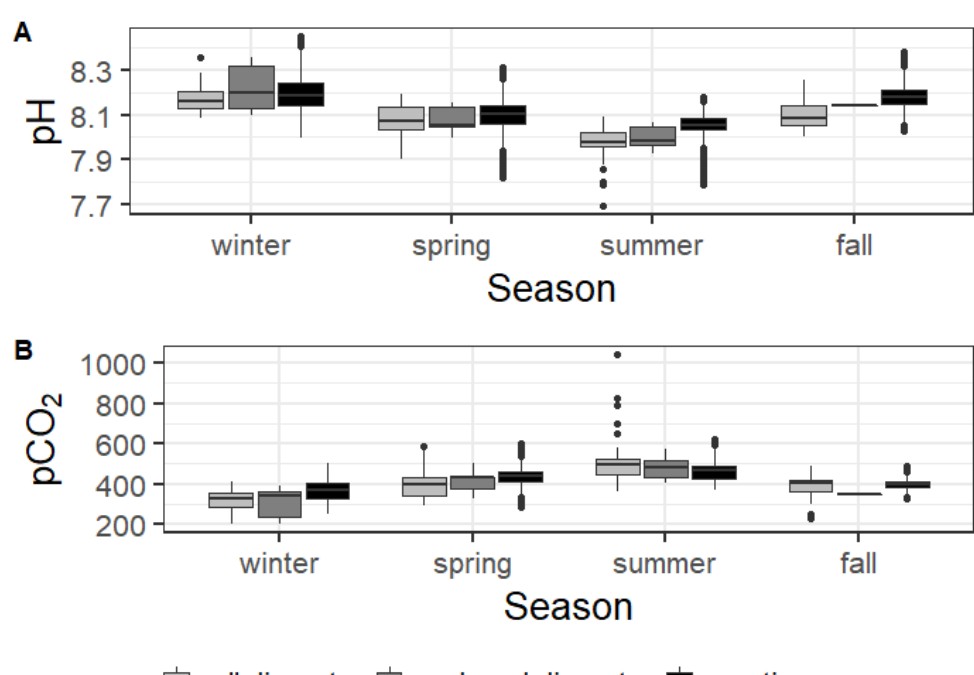

**Figure 3.** Boxplots of seasonal variability in pH and $p$CO$_2$ using all discrete data (May 2,
2014- Feb. 25, 2020), reduced discrete data (Nov. 8 2016 – Aug 23, 2017, to overlap with
continuous monitoring), and continuous sensor data (Nov. 8 2016 – Aug 23, 2017)
Table 2. Diel variability in system parameters from continuous sensor data (Nov 8, 2016
– Aug 23, 2017). The p-values reported are from a paired-$t$ test comparing the means of
each day (9am-3pm LST) with the mean of the same night (9pm – 3am LST); all
significant results based on α=0.05 are bolded. Diel range calculations were done using
only days with the full 24 hours of hourly measurements (176 out of 262 measured) to
ensure that data gaps did not influence the calculations. Reported fluxes use the Jiang et
al. (2008) gas transfer velocity parameterization. Note that the Fall season had much
fewer observations than other seasons because of the timing of sensor deployment.

| Parameter | Time Period | Daytime Mean | Nighttime Mean | Day versus Night p-value | Mean Diel Range | Minimum Diel Range | Maximum Diel Range |
|---|---|---|---|---|---|---|---|
| **Temperature (°C)** | **Full Sampling Period** | $23.0 \pm 5.3$ | $23.2 \pm 5.4$ | **<0.0001** | $1.3 \pm 0.8$ | 0.30 | 3.93 |
| | Winter | $17.2 \pm 2.1$ | $17.4 \pm 2.1$ | 0.2055 | $1.5 \pm 0.8$ | 0.3 | 3.8 |
| | Spring | $23.7 \pm 2.7$ | $23.8 \pm 2.9$ | 0.5579 | $1.2 \pm 0.6$ | 0.3 | 3.0 |
| | Summer | $29.6 \pm 0.7$ | $29.9 \pm 0.8$ | **<0.0001** | $1.0 \pm 0.6$ | 0.3 | 3.8 |
| | Fall | $22.0 \pm 1.19$ | $23.0 \pm 1.0$ | **<0.0001** | $1.8 \pm 0.9$ | 0.8 | 3.9 |



| **Salinity** | **Full Sampling Period** | $30.5 \pm 4.1$ | $31.0 \pm 3.3$ | **0.0004** | $3.4 \pm 2.7$ | 0.250 | 15.870 |
|---|---|---|---|---|---|---|---|
| | Winter | $29.6 \pm 4.2$ | $30.4 \pm 3.1$ | **0.0051** | $3.8 \pm 2.2$ | 0.25 | 9.48 |
| | Spring | $30.1 \pm 2.6$ | $30.2 \pm 2.6$ | 0.5604 | $2.5 \pm 2$ | 0.4 | 8.17 |
| | Summer | $33.4 \pm 3.2$ | $33.1 \pm 3.3$ | 0.0550 | $2.0 \pm 1.7$ | 0.3 | 9.73 |
| | Fall | $25.9 \pm 3.9$ | $29.0 \pm 3.2$ | **<0.0001** | $7.7 \pm 3.6$ | 1.2 | 15.87 |
| **pH** | **Full Sampling Period** | $8.12 \pm 0.10$ | $8.13 \pm 0.09$ | **<0.0001** | $0.09 \pm 0.05$ | 0.02 | 0.28 |
| | Winter | $8.18 \pm 0.08$ | $8.20 \pm 0.07$ | **0.0108** | $0.10 \pm 0.05$ | 0.02 | 0.28 |
| | Spring | $8.09 \pm 0.09$ | $8.10 \pm 0.08$ | 0.3286 | $0.08 \pm 0.03$ | 0.03 | 0.18 |
| | Summer | $8.04 \pm 0.06$ | $8.07 \pm 0.05$ | **<0.0001** | $0.08 \pm 0.04$ | 0.03 | 0.19 |
| | Fall | $8.20 \pm 0.05$ | $8.17 \pm 0.05$ | **0.0038** | $0.12 \pm 0.04$ | 0.03 | 0.20 |
| **$p\mathrm{CO_2}$ (µatm)** | **Full Sampling Period** | $417 \pm 54$ | $416 \pm 65$ | 0.7065 | $58 \pm 33$ | 12.6 | 211.3 |
| | Winter | $374 \pm 44$ | $358 \pm 43$ | **<0.0001** | $43 \pm 21$ | 12.6 | 121.1 |
| | Spring | $438 \pm 42$ | $437 \pm 48$ | 0.7237 | $61 \pm 31$ | 20.5 | 152.8 |
| | Summer | $452 \pm 44$ | $471 \pm 51$ | **0.0003** | $74 \pm 42$ | 23.6 | 211.3 |
| | Fall | $406 \pm 24$ | $399 \pm 27$ | 0.0545 | $56 \pm 18$ | 22 | 92.2 |
| **$\mathrm{CO_2}$ Flux (mmol m$^{-2}$ d$^{-1}$)** | **Full Sampling Period** | $0.0 \pm 6.3$ | $-1.3 \pm 5.9$ | 0.3028 | $34.1 \pm 29.0$ | 2.7 | 189.0 |
| | Winter | $-14.9 \pm 8.4$ | $-19.1 \pm 7.7$ | 0.0676 | $46.6 \pm 38.9$ | 2.7 | 189.0 |
| | Spring | $7.6 \pm 5.2$ | $7.0 \pm 5.2$ | 0.6680 | $27.5 \pm 18.5$ | 4.9 | 115.0 |
| | Summer | $9.4 \pm 5.6$ | $11.7 \pm 5.2$ | 0.1167 | $32.3 \pm 22.9$ | 4.5 | 111.0 |
| | Fall | $0.1 \pm 3.8$ | $-0.3 \pm 3.5$ | 0.7449 | $17.0 \pm 10.2$ | 3.9 | 40.1 |


Table 3. Tests examining differences in mean carbonate system parameters between seasons and between types of sampling (continuous monitoring with sensors Nov. 8 2016 – Aug 23, 2017, discrete sample collection and laboratory measurement during only the continuous monitoring period Nov. 8 2016 – Aug 23, 2017, and discrete sample collection and laboratory measurement during the entire sampling period May 2, 2014- Feb. 25, 2020). For both the two-way ANOVA and associated one-way ANOVAs, p-values are listed. All significant results based on α=0.05 are bolded, and the F statistic is in parentheses. Since all two-way ANOVAs had a significant interaction between factors, individual one-way ANOVAs were conducted for each level of the other factor. Following significant one-way ANOVAs, multiple comparisons using the Westfall adjustment (Westfall, 1997) were conducted; individual comparisons with significantly different means (based on α=0.05) are listed as unequal beneath the one-way ANOVA results (All≠ indicates that every individual comparison between levels had significantly different means. W = winter, Sp = spring, Su = summer, F = fall; C = continuous sensor data, D = discrete sample data over the entire discrete monitoring period, D_C = Discrete sample data during only the period of continuous monitoring).

| **Parameter** | **Two-way ANOVA** | | | **One-way ANOVA and post-hoc multiple comparison results for differences between types of sampling** | | | | **One-way ANOVA and post-hoc multiple comparison results for difference between seasons** | | |
|---|---|---|---|---|---|---|---|---|---|---|
| | Interaction | Season | Sampling type | winter | spring | summer | fall | Continuous | Discrete (Continuous Period) | Discrete (Entire Period) |





| | | | | | | | | | | |
|---|---|---|---|---|---|---|---|---|---|---|
| **Temp (°C)** | **<0.0001** (15.8) | **<0.0001** (12369.7) | 0.7346 (0.3) | 0.0710 (2.6) | 0.1052 (2.3) | **<0.0001** (19.6)<br>D≠C | **<0.0001** (61.4)<br>D≠C | **<0.0001** (12559)<br>All≠ | **<0.0001** (22.8)<br>W≠Su;<br>W≠Sp;<br>W≠F;<br>Su≠Sp;<br>Su≠F | **<0.0001** (58.2)<br>All≠ |
| **Salinity** | **0.0141** (2.7) | **<0.0001** (598.7) | 0.6509 (0.4) | 0.1716 (1.8) | **0.0013** (6.7)<br>D≠C | 0.1921 (1.7) | 0.7007 (0.4) | **<0.0001** (580.0)<br>W≠Su;<br>W≠F;<br>Su≠Sp;<br>Su≠F;<br>Sp≠F | 0.2516 (1.6) | **<0.0001** (17.5)<br>W≠Su;<br>Su≠Sp;<br>Su≠F |
| **pH** | **0.0013** (3.7) | **<0.0001** (1412.3) | **<0.0001** (24.0) | 0.4026 (0.9) | 0.9238 (0.1) | **<0.0001** (24.1)<br>D≠C<br>**C≠D$_C$** | **<0.0001** (33.2)<br>D≠C | **<0.0001** (1381.2)<br>All≠ | **0.0152** (5.7)<br>W≠Su | **<0.0001** (35.3)<br>W≠Su;<br>W≠Sp;<br>W≠F;<br>Su≠Sp;<br>Su≠F |
| **$pCO_2$ (µatm)** | **<0.0001** (10.4) | **<0.0001** (1747.3) | **0.0147** (4.2)<br>D≠C | **0.0018** (6.4)<br>D≠C | **<0.0001** (17.4)<br>D≠C | **0.0002** (8.4)<br>D≠C | 0.0398 (3.2)<br>**All=** | **<0.0001** (1737.6)<br>All≠ | **0.0407** (4.0)<br>W≠Su | **<0.0001** (8.4)<br>W≠Su;<br>W≠Sp;<br>W≠F;<br>Su≠Sp;<br>Su≠F |
| **$CO_2$ Flux (mmol m$^{-2}$ d$^{-1}$)** | **0.0144** (2.6) | **<0.0001** (738.1) | 0.6739 (0.4) | 0.9140 (0.1) | **<0.0001** (11.8)<br>D≠C | **0.0214** (3.9)<br>D≠C | 0.5849 (0.5) | **<0.0001** (725.9)<br>All≠ | **0.0299** (4.5)<br>W≠Su | **<0.0001** (19.2)<br>W≠Su;<br>W≠F;<br>Su≠Sp;<br>Su≠F |


There was substantial diel variability in parameters (Table 2, Fig. 4). Over the 10-
month in-situ monitoring period, temperature had a mean diel range (daily maximum
minus daily minimum) of 1.3 ± 0.8°C (Table 2). Daytime and nighttime temperature
differed significantly during the summer and fall months, with higher temperatures at
night for both seasons (Table 2). The mean diel range of salinity was 3.4 ± 2.7 (Table 2).
Daytime and nighttime salinity differed significantly during the winter and fall months,
with higher salinities at night for both seasons. The mean diel range of pH was 0.09 ±



0.05 (Table 2). Daytime and nighttime pH differed significantly during the winter,
summer, and fall months; nighttime pH was significantly higher than that of the daytime
during the summer and winter months, and daytime pH was significantly higher during
the fall (Table 2, Fig. 4). The mean diel range of $p\mathrm{CO_2}$ was $58 \pm 33$ µatm (Table 2, Fig.
4). Daytime and nighttime $p\mathrm{CO_2}$ differed significantly during the winter and summer
months; nighttime $p\mathrm{CO_2}$ was significantly higher than that of the daytime during the
summer and daytime $p\mathrm{CO_2}$ was significantly higher during the winter (Table 2, Fig. 4).

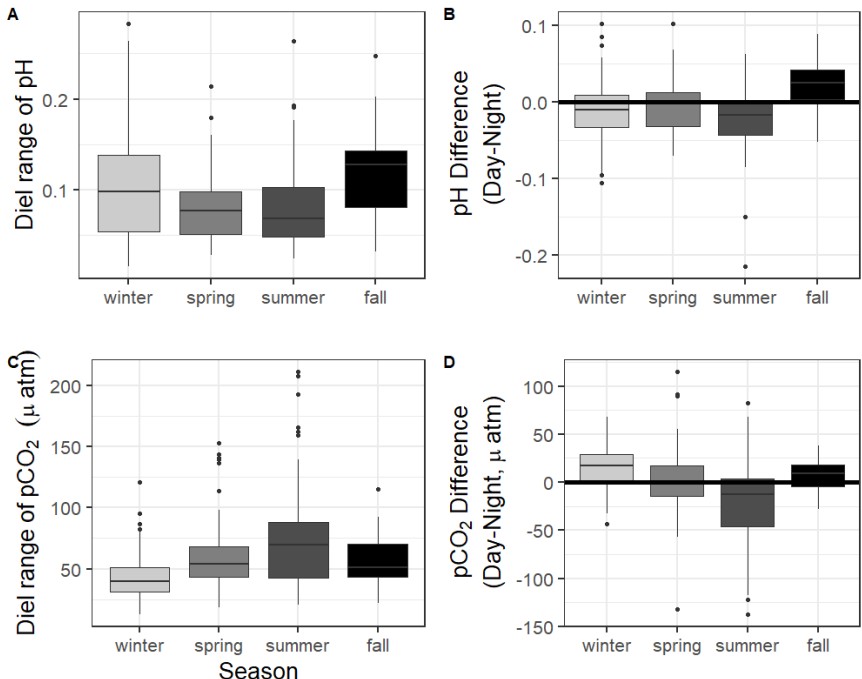

Figure 4. Boxplots showing the diel range (maximum minus minimum) and difference in
daily parameter mean daytime minus nighttime measurements for pH and $p\mathrm{CO_2}$ from
continuous sensor data.

The seasonally weighted mean $\mathrm{CO_2}$ flux calculated from sensor data across the

entire monitoring period was $0.2 \pm 23.7$ mmol m$^{-2}$ d$^{-1}$ (Table 1). Mean $\mathrm{CO_2}$ flux differed
by season (Table 3). Winter and fall both had net negative $\mathrm{CO_2}$ flux (winter was most



negative), and summer and spring both had a net positive $CO_2$ flux (summer was most
positive) (Table 1, Fig. 5). $CO_2$ flux also fluctuated on a daily scale, with the mean diel
range (daily maximum – minimum) over the entire monitoring period being $34.1 \pm 29.0$
mmol m$^{-2}$ d$^{-1}$ (Table 2). However, there was not a significant difference in $CO_2$ flux
calculated for daytime versus nighttime hours for the entire monitoring period or any
individual season based on $\alpha=0.05$ (paired t-test, Table 2).

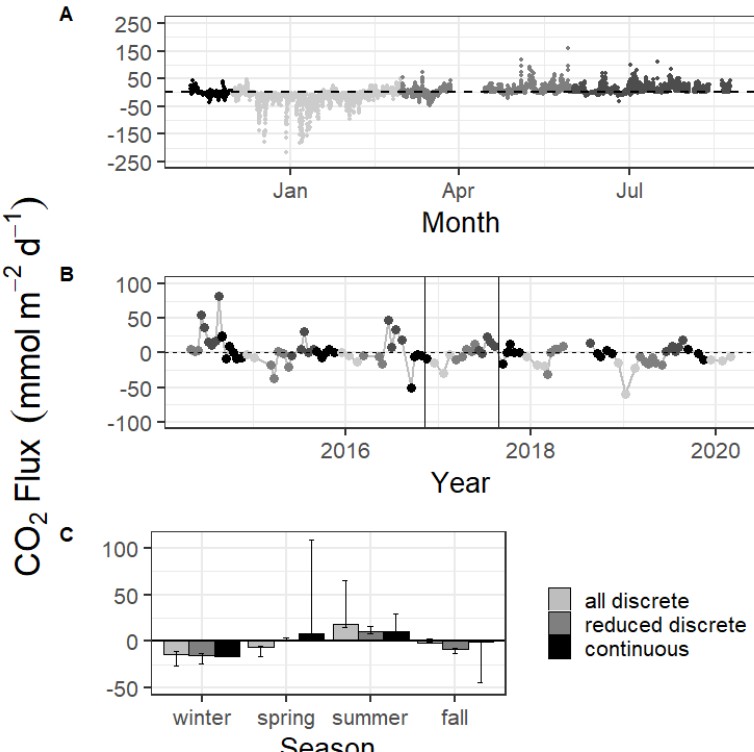

**Figure 5.** $CO_2$ flux calculated over the sampling periods from continuous (A) and
discrete (B) data using the Jiang et al. (2008) wind speed parameterization. Gray scale in
(A) and (B) denote different seasons. Vertical lines in (B) denote the time period of
continuous monitoring.  (C) shows the seasonal mean $CO_2$ flux calculated using the Jiang
et al. (2008) gas transfer velocity parameterization and error bars representing mean $CO_2$
flux calculation using Ho and Raymond and Cole windspeed parameterizations. The
different color bars within each season represent all discrete data (May 2, 2014- Feb. 25,
2020), reduced discrete data (Nov. 8 2016 – Aug 23, 2017, to overlap with continuous
monitoring), and continuous sensor data (Nov. 8 2016 – Aug 23, 2017).



The relative influence of thermal and nonthermal factors (T/B) in controlling
$p$CO$_2$ varied over different time scales (Table 4, Fig. 6). T/B calculated from sensor data
for the entire period was 0.98, indicating that the magnitude of control of non-thermal
processes on $p$CO$_2$ was slightly greater than that of temperature. Within seasons, T/B
calculated from sensor data ranged from 0.51 in the winter to 0.69 in the spring, showing
that non-thermal processes exert more control on $p$CO$_2$ within each individual season
(Table 4). On a daily scale, only 11 of the 178 days with measurements for all 24 hours
had temperature control of $p$CO$_2$ exceeding the non-thermal control (Table 4, Fig. 6).
**Table 4.** Thermal versus non-thermal control on $p$CO$_2$ (Takahashi et al. 2002) over
different time scales using both continuous sensor data and discrete sample data
(indicated as Sampling Type C and D, respectively). If more than one segment of time is
being considered (n>1), $\Delta p$CO$_2$ values are the mean ± standard deviation of all segments,
the T/B values are the minimum and maximum T/B, and the number out of n with T/B>1
(indicating greater control of $p$CO$_2$ by temperature than other processes) is recorded. The
summary of annual T/B values from discrete data includes only 2015-2019 (n=5 years;
2014 and 2020 were omitted since monitoring did not occur throughout the entire year).
Daily values from continuous data were only reported for those days with all 24
measurements.

| Time Period / Scale | Sampling type | n | $\Delta p$CO$_{2, thermal}$ (µatm) | $\Delta p$CO$_{2, nonthermal}$ (µatm) | T/B | Number out of n with T/B >1 |
|---|---|---|---|---|---|---|
| Full Monitoring Period (May 2, 2014- Feb. 25, 2020) | D | 1 | 301.9 | 537.8 | 0.56 | |
| Annual | D | 5 | 259.3 ± 16.0 | 319.1 ± 130.9 | 0.48 – 1.17 | 2/5 |
| Continuous Monitoring Period | C | 1 | 355.0 | 360.7 | 0.98 | |
| (Nov 2016 – August 2017) | D | 1 | 236.3 | 229.9 | 1.03 | |
| Winter | C | 1 | 168.2 | 328.4 | 0.51 | |
| | D | 6 | 42.2 ± 23.4 | 101.7 ± 78.7 | 0.20 – 4.90 | 1/6 |
| Spring | C | 1 | 171.4 | 246.9 | 0.69 | |
| | D | 6 | 142.3 ± 53.7 | 147.8 ± 67.3 | 0.59 – 2.42 | 2/6 |
| Summer | C | 1 | 100.2 | 179.9 | 0.56 | |
| | D | 6 | 46.9 ± 26.6 | 176.9 ± 108.3 | 0.21 – 0.35 | 0/6 |
| Fall | C | 1 | 105.9 | 181.6 | 0.58 | |
| | D | 6 | 179.8 ± 59.5 | 176.6 ± 78.1 | 0.59 – 3.06 | 2/6 |
| Daily | C | 178 | 21.8 ± 11.8 | 63.8 ± 30.3 | 0.05 – 1.68 | 11/178 |






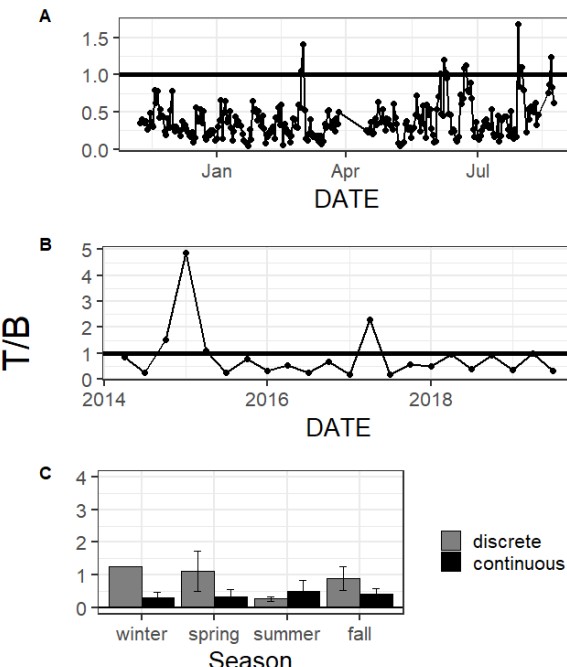

**Figure 6.** T/B (thermal $p$CO$_2$/ non-thermal $p$CO$_2$) calculated for each day from
continuous (A) and each season from discrete (B) data. Bar graphs showing the seasonal
mean and standard deviation of T/B from both discrete and continuous data (C).

Tidal fluctuations had a significant effect on carbonate system parameters (Table

5). Both temperature and salinity were higher at low tide during the winter and summer
months and higher at high tide during the spring. pH was higher at high tide during the
winter and summer and higher at low tide during the spring, and $p$CO$_2$ was higher during
low tide during winter, spring, and summer (Table 5). CO$_2$ flux also varied with tidal
fluctuations. CO$_2$ flux was higher in the low tide condition for all season with tide data;
the location was less of a CO$_2$ sink during low tide conditions in the winter and more of a
CO$_2$ source during low tide conditions in the spring and summer.
**Table 5.** Differences in temperature, salinity and mean carbonate system parameters from
continuous sensor data between high tide and low tide. High tide was defined as a tide
level greater than Q3 and low tide was defined as a tide level less than Q1. Seasons were
examined separately with t-tests because of a significant interaction (based on α=0.05)



between the season and high/low tide factors in a two-way ANOVA. Fall was omitted
from the analysis because tide data was only available at the location beginning
December 20, 2016.

| Parameter | Season | High Tide Mean | Low Tide Mean | Difference between tide levels, t-test p-value |
|---|---|---|---|---|
| **Temperature (°C)** | Winter | $16.7 \pm 1.7$ | $17.6 \pm 2.0$ | **<0.0001** |
| | Spring | $24.4 \pm 2.7$ | $23.6 \pm 2.7$ | **<0.0001** |
| | Summer | $29.3 \pm 0.5$ | $30.1 \pm 0.7$ | **<0.0001** |
| **Salinity** | Winter | $30.2 \pm 2.5$ | $31.3 \pm 2.9$ | **<0.0001** |
| | Spring | $30.4 \pm 1.9$ | $30.0 \pm 2.7$ | **0.0071** |
| | Summer | $30.5 \pm 2.4$ | $34.5 \pm 3.0$ | **<0.0001** |
| **pH** | Winter | $8.20 \pm 0.08$ | $8.15 \pm 0.06$ | **<0.0001** |
| | Spring | $8.07 \pm 0.09$ | $8.10 \pm 0.07$ | **<0.0001** |
| | Summer | $8.08 \pm 0.04$ | $8.04 \pm 0.06$ | **<0.0001** |
| **$p\mathrm{CO_2}$ (µatm)** | Winter | $331 \pm 40$ | $378 \pm 42$ | **<0.0001** |
| | Spring | $435 \pm 33$ | $443 \pm 50$ | **0.0154** |
| | Summer | $419 \pm 30$ | $482 \pm 48$ | **<0.0001** |
| **$\mathrm{CO_2}$ Flux (mmol m$^{-2}$ d$^{-1}$)** | Winter | $-33.0 \pm 38.1$ | $-11.7 \pm 21.8$ | **<0.0001** |
| | Spring | $7.4 \pm 14.0$ | $8.7 \pm 14.8$ | **0.2248** |
| | Summer | $1.8 \pm 6.3$ | $16.0 \pm 14.5$ | **<0.0001** |


*3.2 Discrete sampling results*
All results reported here are for the entire 5+ years of monitoring; the subset of
discrete sample data that overlaps with the continuous monitoring period will be
addressed only in the discussion for method comparisons. All reported discrete sampling
parameters showed substantial temporal variability over the 5+ years of monitoring (Fig.
2E-H). The mean temperature was $24.1 \pm 5.3$°C, ranging from $11.8 - 31.2$°C; the mean
salinity was $30.1 \pm 4.4$, ranging from $16.7 - 37.5$; the mean pH was $8.079 \pm 0.092$,
ranging from 7.693 to 8.354; and the mean $p\mathrm{CO_2}$ was $406 \pm 100$ µatm, ranging from 199
to 1043 (Table 1). These parameters all experienced significant seasonal variability
(Tables 1 and 3). Temperature was significantly different between each season, highest in
summer and lowest in winter (Tables 1 and 3). Salinity was highest during the summer
months and was not significantly different between other seasons (Tables 1 and 3). pH
and $p\mathrm{CO_2}$ were both significantly different between all seasons with the exception of



spring and fall (Table 3). Winter had both the highest seasonal pH ($8.162 \pm 0.065$) and
lowest seasonal $p\mathrm{CO}_2$ ($331 \pm 39$ µatm), and summer had both the lowest seasonal pH
($7.975 \pm 0.046$) and highest seasonal $p\mathrm{CO}_2$ ($511 \pm 108$) (Tables 1 and 3, Fig. 3).

Average annual $\mathrm{CO}_2$ flux calculated with discrete sample data was slightly

negative ($-0.9 \pm 18.7$ mmol m$^{-2}$ d$^{-1}$, Table 1). $\mathrm{CO}_2$ flux varied greatly by season. Summer,
the only season with a net positive $\mathrm{CO}_2$ flux over the 5+ year period, had significantly
higher flux than all other seasons; winter had the lowest calculated flux, but it was not
significantly different from spring (Tables 1 and 3).

As with the continuous data, T/B calculated from the discrete data varied over

different time scales (Table 4, Fig. 6). For the entire period, T/B was 0.56, indicating that
non-thermal processes exerted more control than temperature on $p\mathrm{CO}_2$. The annual T/B
varied from 0.48 to 1.17, with two of the five sampled years having T/B greater than one
(i.e. more thermal influence). While the majority of individual seasons that were sampled
experienced stronger non-thermal control on $p\mathrm{CO}_2$ (T/B $<$1), the only season that never
experienced stronger thermal control was summer, with summer T/B values ranging from
$0.21 - 0.35$ for the 6 sampled years (Table 4).
**Discussion**
*4.1 Factors controlling temporal variability in carbonate system parameters*
*4.1.1 Thermal versus non-thermal control of pCO$_2$*

Substantial variability in the carbonate system was observed at the study site over

multiple time scales including diel, seasonal, and interannual. Many physical and
biological factors (e.g., temperature, currents, tides, wind speed, net ecosystem
metabolism, etc.) can exert control on $p\mathrm{CO}_2$ and subsequently exert control on other



carbonate system parameters. Using the thermal versus non-thermal analysis of control
on $p$CO$_2$ from Takahashi et al. (2002), we were able to determine that non-thermal
processes generally exert more control on the $p$CO$_2$ in the Aransas Ship Channel relative
to temperature over multiple time scales (Table 4, T/B<1). Only five of 24 seasons (one
winter, two spring, and two fall) throughout the years of discrete sampling had greater
variability in $p$CO$_2$ attributed to temperature ($\Delta p$CO$_{2,\ thermal}$) than other processes ($\Delta p$CO$_{2,}$
$_{nonthermal}$) (Table 4). The magnitude of $p$CO$_2$ variation attributed to non-thermal processes
varied greatly over multiple time scales (i.e. $\Delta p$CO$_{2,\ nonthermal}$ had large standard
deviations, Table 4). For example, in 2016 $p$CO$_2$ had the strongest non-thermal control of
any year, with a $\Delta p$CO$_{2,\ nonthermal}$ of 538 µatm, while 2019 had the weakest control from
non-thermal processes of any year, with a $\Delta p$CO$_{2,\ nonthermal}$ of 208. Conversely, the
magnitude of $p$CO$_2$ variation attributed to temperature was consistent across time scales.
For example, in 2015 $p$CO$_2$ had the strongest thermal control of any year, with a $\Delta p$CO$_{2,}$
$_{thermal}$ of 276 µatm, while 2019 had the weakest thermal control of any year, with a
$\Delta p$CO$_{2,\ thermal}$ of 243 µatm.

The difference in T/B between sampling methods is relatively small over the 10-

month sensor deployment period (Table 4). Each method suggested temperature and
nonthermal processes exert a relatively similar control on $p$CO$_2$, but continuous
monitoring demonstrated a greater magnitude of fluctuation resulting from both
temperature and non-thermal processes (i.e. greater    $\Delta p$CO$_{2,\ thermal}$ and $\Delta p$CO$_{2,\ nonthermal}$).
Over shorter time scales, like individual seasons, the calculated T/B did not align well
between sampling methods. The T/B (calculated using only data from Nov 2016 –
August 2017) for discrete versus continuous sampling was respectively 0.17 versus 0.51





for winter, 2.37 versus 0.69 for spring, and 0.20 versus 0.56 for summer (Table 4, Fall
was omitted since there was only one discrete sampling event during the fall of the
continuous monitoring period). Sampling bias due to the small number of within-season
sampling events for the discrete monitoring likely resulted in this difference. From both
continuous and discrete data, summers always had stronger control exerted on $pCO_2$ from
nonthermal processes than temperature. While temperatures were high during the
summer months, the within-season variability in temperature was the lowest (Table 1);
less of a temperature swing resulted in less thermal control on the system. Conversely,
spring and fall seasons, which experienced the greatest temperature swings (Table 1), had
greater relative temperature control exerted on $pCO_2$ (Table 4). The differences in
$\Delta pCO_{2, \text{thermal}}$ and $\Delta pCO_{2, \text{nonthermal}}$ between monitoring methods illustrate that there is
information that is missed when only sampling bimonthly/monthly and during the
daytime. Generally, both $\Delta pCO_{2, \text{thermal}}$ and $\Delta pCO_{2, \text{nonthermal}}$ are higher when calculated
from sensors than discrete sampling, indicating that the extremes are generally not
captured by the discrete sampling and sensor data would provide a better understanding
of system controls.

The relative importance of thermal versus non-thermal controls may be modulated

by tide level. The influence of tides can be removed from the calculated non-thermal
$pCO_2$ term, leaving only biological processes and other physical controls in the non-
thermal term, by examining periods of high tide and low tide separately. Using our sensor
data and the same water level data used for the tide analysis, we found that T/B is higher
during the high tide condition within each season. T/B for high tide and low tide,
respectively, was 0.60 and 0.52 for winter, 0.84 and 0.66 for spring, and 0.62 and 0.58 for



summer. The higher control exerted by nonthermal processes during low tide seems
intuitive given that there is less volume of water for the end products of biological
processes to build up in. The difference in T/B between high tide and low tide conditions
was greatest in the spring, likely due to a combination of elevated spring-time
productivity and larger tidal ranges in the spring.

Using data from the first year of our discrete sampling (May 2014 – April 2015),

Yao and Hu (2017) reported that the Aransas Ship Channel T/B was 1.53 during drought
and 1.79 during a period of flooding, both of which are significantly higher than what we
found over most timescales (the exception being certain individual seasons, mostly
during that first year of sampling, Table 4, Fig. 6B). Yao and Hu (2017) also found that
locations in the upper estuary experienced lower T/B during flooding conditions than
drought conditions, but the opposite was found for the Aransas Ship Channel location,
where the flooding conditions had higher T/B. It is likely that the high T/B calculated by
Yao and Hu (2017) was a result of the drought condition at the beginning of their
sampling; given the long residence time of MAE, the Aransas Ship Channel may not
have experienced the influence of the freshwater inflow by the end of the Yao and Hu
(2017) study. Once the freshwater reached the Aransas Ship Channel location, it would
likely experience a reduced T/B as did the upper parts of the system. Since then, there has
not been another significant drought in the system, so it seems as though the non-thermal
controls on $p$CO$_2$ are more important at this location under normal freshwater inflow
conditions.



*4.1.2 Investigating controls on the carbonate system using relationships between*
*carbonate system parameters and other environmental parameters*

We further investigated controls on the carbonate system using tide and

windspeed data (obtained from NOAA's Aransas Pass station at
https://tidesandcurrents.noaa.gov/) and dissolved oxygen, PAR, turbidity, and chlorophyll
fluorescence data (obtained from the MANERR at
https://missionaransas.org/science/download-data) along with our continuous and discrete
data. All investigations of relationships between environmental parameters discussed
below included only the observations with no significant water column stratification
(defined as a salinity difference of less than 3 between surface water from our YSI and
bottom water (>5 m) from the MANERR's YSI). This omission of stratified water was
intended to omit instances of substantial differences in chemical parameters between the
surface and bottom water since all MANERR environmental data used in our analysis
were measured at depth while our sensors measured surface water. Omitting stratified
water reduced our continuous dataset from 6088 to 5524 observations, and omitting
observations where there were no MANERR data to determine stratification further
reduced the dataset to 4112 observations. Similarly, removing instances of stratification
reduced discrete sample data from 104 to 89 surface water observations.

To extend upon the above discussion of thermal versus non-thermal controls on

$p\mathrm{CO_2}$, the extent of thermal control on both pH and $p\mathrm{CO_2}$ can be investigated based on
relationships between parameters. There is a strong negative correlation between pH and
temperature and a strong positive correlation between $p\mathrm{CO_2}$ and temperature (Table 6,
Fig. 7). The direction of these relationships (sign of the correlation coefficient) at the



Aransas Ship Channel was the same as in open ocean waters despite these relationships
not being consistent across different estuarine environments (N. Rosenau, personal
communications). The strong correlations with temperature support our findings that
thermal controls on $p\text{CO}_2$ can be important over multiple time scales. Significantly
warmer water temperatures were observed during the nighttime in both summer and fall
(Table 2, Fig. 8), indicating that temperature could exert a slight control on the carbonate
system over a diel time scale. More substantial temperature swings between seasons
indicate that temperature is more important over seasonal time scales (Table 1). In
addition to direct thermal control at our site, the strong correlations with temperature are
likely derived from changes in net community metabolism associated with temperature
(Caffrey, 2004). For example, the strong negative correlation between nonthermal $p\text{CO}_2$
and temperature (Table 6) is likely indicative of enhanced primary productivity in
warmer waters.

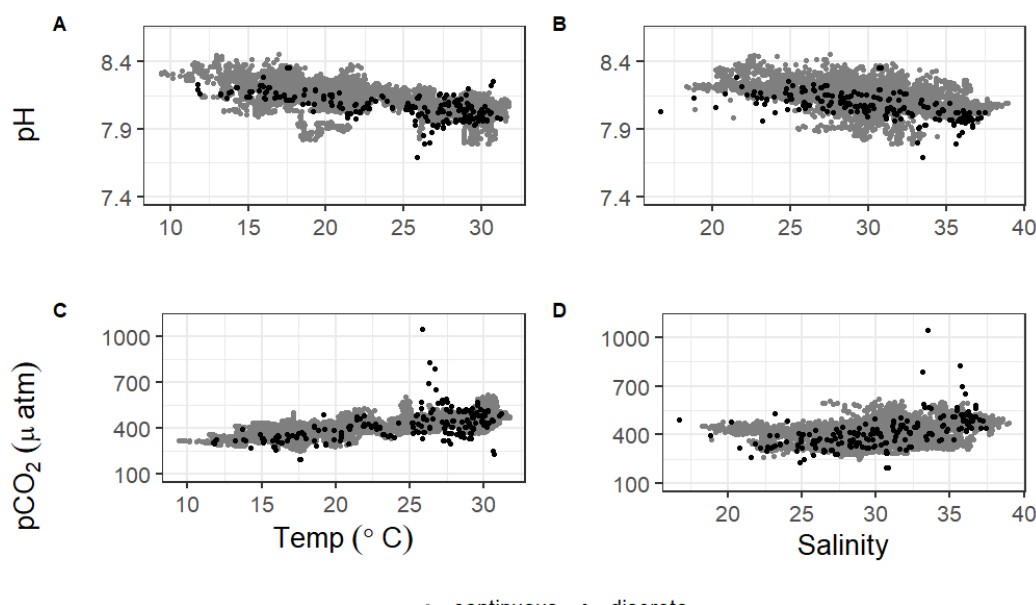




Figure 7. Correlations of pH and $p$CO$_2$ with temperature and salinity from continuous
sensor data (gray) and all discrete data (black).

Table 6. Pearson correlation coefficients between surface water carbonate system
parameters and other water quality and environmental parameters for both continuous
sensor data and discrete sample data (entire sampling period). Only observations without
significant stratification in the water column were included in these analyses. Parameter
pairs with a significant correlation based on α=0.05 have a correlation coefficient
reported. Asterixis are used to indicate the level of significance of the correlation, *
$p<0.05$, ** $p<0.01$, *** $p<0.0001$. The correlation coefficient is listed as 0 if the
relationship was not significant. N/A is listed when the analysis was omitted because the
environmental parameter did not have observations corresponding to the date and time of
at least half of our discrete sample measurements (45 observations).

|  | pH | | $p$CO$_2$ | | $p$CO$_2$, nonthermal | |
|---|---|---|---|---|---|---|
|  | **Continuous** | **Discrete** | **Continuous** | **Discrete** | **Continuous** | **Discrete** |
| Temperature (°C) | -0.55 *** | -0.59 *** | 0.75 *** | 0.53 *** | -0.73 *** | -0.45 *** |
| Salinity | -0.47 *** | -0.74 *** | 0.53 *** | 0.69 *** | -0.28 *** | 0.35 ** |
| Wind Speed (m s$^{-1}$) | -0.04 ** | N/A | 0.15 *** | N/A | 0 | N/A |
| Dissolved Oxygen (mg L$^{-1}$) | 0.55 *** | 0 | -0.81 *** | 0 | 0.45 *** | 0 |
| Tide Level (m) | 0 | 0 | -0.15 *** | 0 | -0.15 *** | -0.55 ** |
| Turbidity | -0.08 *** | N/A | -0.14 *** | N/A | -0.28 *** | N/A |
| Fluor. Chlorophyll | 0.12 *** | N/A | -0.22 *** | N/A | 0.34 *** | N/A |


Though annual average $p$CO$_2$ and CO$_2$ flux are higher in the upper estuary and

lower offshore than at our study site, the same seasonal pattern of elevated $p$CO$_2$ and
positive CO$_2$ flux in the summer and depressed $p$CO$_2$ and negative CO$_2$ flux during the
winter observed at our site has also been observed throughout the entire MAE and in the
open Gulf of Mexico (Hu et al., 2018; Yao and Hu, 2017). Seasonal fluctuations in pH
and $p$CO$_2$ are low at our study site relative to other systems that have been studied to date
(Carstensen et al., 2018; Yao and Hu, 2017), which may be in part due to the relatively
small seasonal temperature changes (Table 1) in this warm, semiarid environment.
Despite substantial seasonal thermal control at our site, simple linear regressions indicate
that temperature had substantially higher explanatory value for pH and $p$CO$_2$ in offshore
GOM waters ($R^2$ = 0.81 and 0.78, respectively (Hu et al., 2018)) than at our site ($R^2$ =



0.30 and 0.52, respectively, for sensor data and $R^2$ = 0.38 and 0.25, respectively, for
discrete data).
Other physical factors that may exert control on the carbonate system (including
windspeed, salinity, tide level, and turbidity) can also be investigated through parameter
relationships. We investigated wind speed as a possible control on the carbonate system
to gain insight into the effect of wind-driven $CO_2$ fluxes on the inventory of $CO_2$ in the
water column (and subsequent impacts to the entire carbonate system). The Texas coast
has relatively high wind speeds, with the mean wind speed observed during our
continuous monitoring period being 5.8 m s$^{-1}$. While this results in relatively high
calculated $CO_2$ fluxes (Fig. 5), the seasonal relationship between $p$CO$_2$ and windspeed
does not support a change in inventory with higher winds. Linear regression analysis
within each season reveals that winter, spring, and fall all experience increases in $p$CO$_2$
with increasing wind, while there is not a significant relationship in summer. Since spring
and summer both have a mean estuarine $p$CO$_2$ greater than atmospheric level (and
positive $CO_2$ flux, Table 1) a negative relationship between windspeed and $p$CO$_2$ would
be necessary to support this hypothesis.
Previous studies have indicated that freshwater inflow may exert a primary
control on the carbonate system in the estuaries of the northwestern GOM (Hu et al.,
2015; Yao et al., 2020; Yao and Hu, 2017). Increased freshwater inflow resulting from
storms has also been shown to increase community respiration, which would
subsequently increase $p$CO$_2$, in the upper reaches of the MAE (Bruesewitz et al., 2013).
MAE is also known to experience large swings in the chemistry of its freshwater inputs,
with relatively high levels of dissolved inorganic carbon and total alkalinity during base





flows but much lower levels due to dilution during intense flooding (Yao et al., 2020).
Given the location of our sampling in the lower portion of the estuary and the long
residence time in the system, we will not directly address freshwater inflows as a
controlling factor, but the influence of freshwater inflow may be evident in the response
of the system to changes in salinity. Carbonate system variability is much lower at our
study site than it is in the more upper reaches of MAE, likely due to the lesser influence
of freshwater inflow and its associated changes in biological activity at the Aransas Ship
Channel (Yao and Hu, 2017). Salinity from both sensor and discrete monitoring was
strongly correlated with pH and $p$CO$_2$, with correlation coefficients nearing (continuous)
or surpassing (discrete) that of the correlations with temperature (Fig. 7; Table 6). Periods
of lower salinity had higher pH and lower $p$CO$_2$, likely due to enhanced freshwater
influence and subsequent elevated primary productivity at the study site. Fluctuating
salinity at the Aransas Ship Channel may also result from direct precipitation,
stratification, and tidal fluctuations. Based on the simple linear regression of salinity with
tide level, there is a significant (p<0.0001) relationship between tide level and salinity,
but the amount of variability in salinity that tides can explain (based on model $R^2$) is only
about 2%.
Tidal fluctuations were clearly important to carbonate system variability at the
Aransas Ship Channel (Table 5). While the northwestern GOM estuaries are generally
microtidal, the constricted tidal inlets such as the Aransas Ship Channel may experience
relatively large tidal fluctuations. The water level data used in this analysis came from a
location directly offshore from our study site, and water level had a range of 1.30 m
(maximum – minimum recorded water level) over the 10-month continuous monitoring



period. Mean water level varied between all seasons; mean spring (highest) water levels
were on average 0.08 m higher than winter (lowest) water levels (ANOVA p<0.0001, fall
was not considered because of a lack of water level data). Tidal influence on pH was less
clear. Data from continuous monitoring did not show a significant correlation between
pH and tide level across the entire monitoring period (Table 6). Significant differences in
mean pH between tide levels were recorded during each season; pH was higher at high
tide (corresponding with the lower $p$CO$_2$) during the winter and summer, but pH was
lower at high tide (conflicting with the lower $p$CO$_2$) in the spring (Table 5). This
separation between water level correlation with pH and $p$CO$_2$ suggests that different
controlling factors of the carbonate system may not be exerted equally on both $p$CO$_2$ and
over different timescales. Similar to pH, both temperature and salinity experienced
seasonally dependent reversals in their difference between tide levels during the spring;
each were higher at low tide during winter and summer and higher at high tide during
spring (Table 5). Given the negative relationship of both temperature and salinity with
pH, it is likely these parameters became important controls on pH in the spring.
To help examine controls on the carbonate system on a diel time scale, we used loess
models (locally weighted polynomial regression) to identify changes in diel patterns over
the course of our monitoring period (Fig. 8). Both tidal and biological controls on the
carbonate system can operate on a diel time scale. The GOM is one of the few places in
the world that experiences diurnal tides (Seim et al., 1987; Thurman, 1994), so
theoretically, the fluctuations in $p$CO$_2$ associated with tides may align to either amplify or
reduce/reverse the fluctuations that would result from diel variability in net community
metabolism. The mean daily tidal fluctuation during our continuous monitoring period



was 0.39 m ± 0.13 m, which did not significantly differ between seasons (ANOVA
p=0.739). However, diel patterns in tidal fluctuations exhibited a strong seasonal pattern
during the continuous monitoring period, with spring and summer having higher tide
level during the daytime and winter having higher tide level during the nighttime (Fig. 8).
This same seasonal pattern in diel tidal fluctuations is exhibited from Dec 20, 2016 (when
the tide data is first available) through the rest of our discrete monitoring period (Feb 25,
2020), indicating that tidal control on diel variability of carbonate system parameters was
likely consistent throughout this time period.

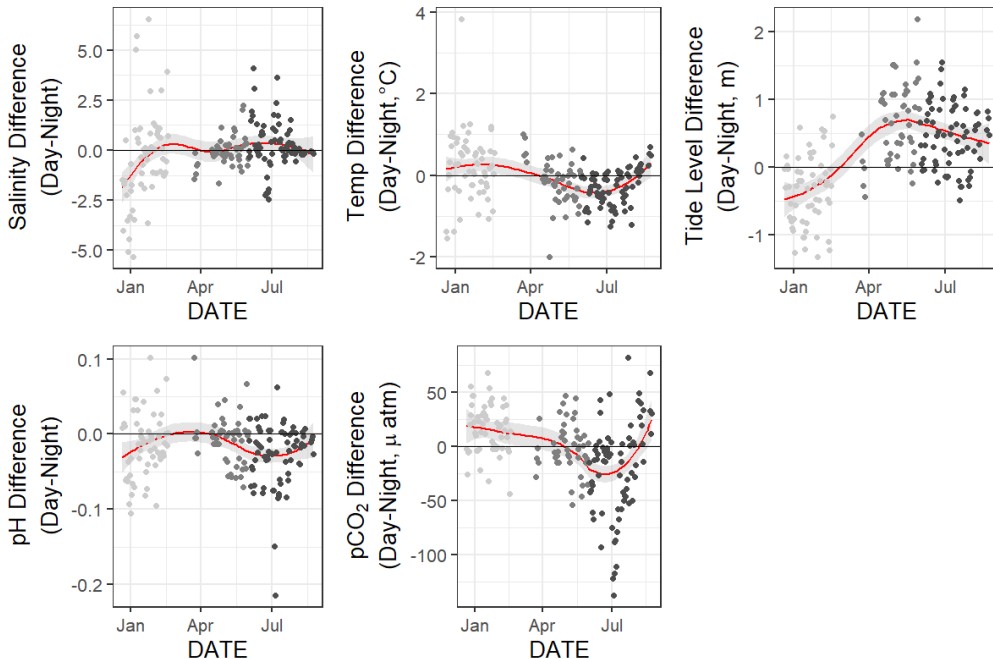

Figure 8. Loess models (red line) and their confidence intervals (gray band around red
line) showing the difference in daily parameter mean daytime minus nighttime
measurements. The gray scale of the data points represents the four seasons over which
data were collected.

Based on diel tidal fluctuations at this site, tidal control should amplify the

biological control signal (nighttime $pCO_2$ > daytime $pCO_2$) during spring and summer





and reduce or reverse the biological control signal during the winter. This was supported
by our $pCO_2$ data, which showed nighttime $pCO_2$ significantly greater than daytime $pCO_2$
in the summer (Table 2). The full reversal of the biological signal in the winter (Table 2,
nighttime $pCO_2$ < daytime $pCO_2$) indicated that biological activity was not the strongest
controlling factor on the diel time scale and was likely exceeded by tidal control. Winter
also had higher daytime temperature (Table 3), which could also contribute to the higher
daytime $pCO_2$, while summer diel temperature and tides would act to amplify the
biological signal.

Again, the diel variability in pH did not mirror $pCO_2$ as would be expected. The

loess models show that daily variability in pH closely mirrors that of temperature while
the daily variability in $pCO_2$ much closer reflects the tide level (Fig. 8), indicating that
controlling factors of the carbonate system may not be exerted equally on both pH and
$pCO_2$.

The extent of biological control on the system can also be investigated based on

correlations between carbonate system parameters and dissolved oxygen (DO).
Respiration-driven acidification is one of the most important local to regional
contributors to acidification in coastal waters, with acidification closely linked to the
widespread issue of deoxygenation (Rabalais et al., 2014; Strong et al., 2014). There were
no observations of hypoxia at our study site during our monitoring, with minimum DO
levels of 3.9 mg L$^{-1}$ and 4.0 mg L$^{-1}$ for our continuous monitoring period and our discrete
sampling period, respectively. Despite the lack of hypoxia, there was a strong
relationship between the carbonate system parameters and DO (Table 6), suggesting that



net ecosystem metabolism may exert an important control on the carbonate system on
certain time scales.

There was no significant difference in daytime and nighttime DO during any

season (paired t-tests, winter p = 0.1573, spring p = 0.4877, summer p = 0.794) despite
the significant differences in pH and $p\mathrm{CO_2}$ between daytime and nighttime (Table 2).
This suggests that net community metabolism is likely not a strong controlling factor of
carbonate system parameters at this site on a diel time scale. The control exerted on the
carbonate system by biological processes is likely much greater on the seasonal scale
than the diel scale. The correlation between continuous $p\mathrm{CO_2}$ and DO is stronger than
$p\mathrm{CO_2}$ and temperature, which suggests strong biological control and supports the
indication by T/B values that non-thermal processes exert more control on $p\mathrm{CO_2}$ than
temperature. Both types of sampling (i.e., continuous and discrete) demonstrate that pH is
generally highest in the winter and lowest in the summer and $p\mathrm{CO_2}$ is highest in the
summer and lowest in the winter (Figs. 2, and 3, Table 1). Though this seasonal pattern
corresponds with the directional response from temperature fluctuations, it can also be
explained by biological activity.

Given that this sampling location is in a ship channel where boat traffic (including

large oil tankers) is relatively heavy, there is potential for atmospheric deposition of acids
(SOx and NOx) to play a role in the carbonate system variability (Doney et al. 2007,
Hunter et al. 2011). To try to understand this control, we deployed air samplers at our
study site for eight 2-week periods. The levels of atmospheric $NO_2$ and $SO_2$ did not vary
widely over the time period. $NO_2$ was ranged 5.45 to 6.99 ppb (6.13 ± 0.63 ppb), and $SO_2$
ranged 1.15 to 1.18 ppb (1.43 ± 0.35 ppb) over the sampling dates (J. D. Felix, personal



communications). There was no apparent correlation between these values and the pH or
$pCO_2$ levels over the 2-week sampling periods.

Co-locating our pH and $pCO_2$ sensors with other coastal environmental

monitoring sensors allowed insight into correlated environmental parameters and
potential driving forces of carbonate chemistry on diel and seasonal time scales. The
results of this study provide strong support for the continued implementation of carbonate
chemistry monitoring in conjunction with preexisting coastal environmental monitoring
infrastructure. Our understanding of any estuarine system could benefit from long-term
effective deployments of these monitoring tools. Strategically locating carbonate
chemistry sensors at estuarine sites that are subject to local OA drivers or support large
biodiversity or commercially important species may be the most crucial in guiding future
mitigation and adaptation strategies for natural systems and aquaculture facilities (Chan
et al., 2013; Strong et al., 2014).
*4.2 Carbonate chemistry as a component of overall estuarine system variability*

Estuaries and coastal areas are dynamic systems with human influence, riverine

influence, and influence from an array of biogeochemical processes, resulting in highly
variable chemical and environmental conditions. To better understand overall system
variability over different time scales, we used a linear discriminant (LD) analysis, a
multivariate statistic that allows dimensional reduction, to determine the linear
combination of environmental parameters (individual parameters reduced into linear
discriminants, LDs) that allow the best differentiation between day and night as well as
between seasons. This used the same suite of environmental data and data sources as
Sect. 4.1.2.





All variables were centered and scaled to allow direct comparison of their
contribution to the system variability. The magnitude (absolute value) of coefficients of
the LDs (Table 7) represents the relative importance of each individual environmental
parameter in the best discrimination between day and night and between seasons, i.e., the
greater the absolute value of the coefficient, the more information the associated
parameter can provide about whether the sample came from day or night (or winter,
spring, or summer). Only one LD could be created for the diel variability (since there are
only two classes to discriminate between – day and night). Two LDs could be created for
the seasonal variability (since there were three classes to discriminate between – fall was
omitted because of the lack of tidal data), but only the coefficients for LD1 are reported
(Table 7) given that LD1 captured 95.64% of the seasonal variability.
Table 7. Coefficients of linear discriminants (LD) from discriminant function analysis
(DFA) using continuous sensor data and other environmental parameters.  Results for
discriminants for both diel and seasonal variability shown. All variables were centered
and scaled. For the seasonal analysis, only LD1 is given since it was captured 95.64% of
the variability (for the diel analysis, there is only one .  Given that many of the water
quality parameters were measured in bottom waters and our sensors were measuring
surface waters, only those observations without significant stratification in the water
column (a salinity difference of less than 3 between surface and bottom) were included in
these analyses.

|  | Diel | Seasonal |
| --- | --- | --- |
|  | LD1 | LD1 |
| Temperature (°C) | 0.5406 | -3.5279 |
| Salinity | 0.1473 | 0.0432 |
| $p\mathrm{CO_2}$ (µatm) | -0.1612 | -0.2928 |
| pH | 0.0593 | 0.0991 |
| Tide Level (m) | 0.0968 | -0.2389 |
| Wind speed (ms$^{-1}$) | -0.0009 | 0.0504 |
| Total PAR | -2.2878 | -0.0676 |
| DO (mg L$^{-1}$) | -0.0839 | 0.0859 |
| Turbidity | -0.0561 | 0.1455 |
| Fluor. Chlorophyll | 0.1397 | -0.4040 |


As would be expected, we found that PAR provided the most differentiation
between daytime and nighttime conditions (based on the largest coefficient associated



with Diel LD1, Table 7). Temperature was the second most important factor in
differentiating between day and night; this corresponds to the diel variability that we
detected where both summer and fall had clear separation of mean temperature between
day and night, with nighttime temperatures being 0.3 and 1.0 higher, respectively (Table
3). The next most important parameter in differentiating between day and night in this
system was $p$CO$_2$, providing more evidence for differentiation between day and night
than other parameters that would be expected to vary on a diel timescale (e.g. chlorophyll
and DO). As for system variability that allowed differentiation between the four seasons,
the most important parameter in system variability was temperature (Table 7, Seasonal
LD1), as would be expected with the clear seasonal temperature fluctuations (Fig. 2E).
The second most important parameter in contributing to seasonal variability was
chlorophyll, likely indicating clear seasonal blooms. The third most important parameter
for seasonal differentiation was $p$CO$_2$; therefore $p$CO$_2$ variability seems to be more
closely tied to seasons than variability in tide level, DO, or the array of other parameters
(Table 7).

The contribution of pH to discriminating along diel or seasonal scales was less

than $p$CO$_2$ despite the same seasonal differences that were identified by ANOVA (Table
3) and more seasons with significant diel differences (Table 2). However, pH still seemed
to be relatively important on seasonal scales, having clearer contribution to seasonal
system variability than several other parameters including DO and salinity.

We can conclude that carbonate chemistry parameters are among the most

important of variants on both daily and seasonal time scales in this coastal setting.
Compared to six other estuaries around the United States with similar sensor deployments



for carbonate chemistry characterization, our study site has a relatively small range of pH
and $pCO_2$ on both diel and seasonal scales (N. Rosenau, personal communication). While
we do not have the same suite of environmental data for these other systems, this
suggests that the relative amount of system variability contributed by carbonate chemistry
may be even greater in other estuarine systems. The relatively small fluctuations in pH
and $pCO_2$ that are seen on a daily scale at the Aransas Ship Channel is likely due to the
subtropical setting with little ocean upwelling influence and the lower estuary position of
our monitoring (further removed from the already small freshwater influence), but it may
also be tied to the system's relatively high buffer capacity. Just as the extent of hypoxia-
induced acidification was relatively low in Corpus Christi Bay compared to other systems
because of the bay's high buffer capacity (McCutcheon et al., 2019), the extent of pH
fluctuation on a daily scale from biological activity would also be modulated by the
intrinsic buffer capacity, which is likely also high in this system due to high alkalinity in
the freshwater endmembers (Yao et al., 2020).
*4.3 Comparing continuous monitoring and discrete sampling*
*4.3.1 Representative sampling in a temporally variable environment*

Discrete water sample collection and analysis is the most common method that

has been employed to attempt to understand the carbonate system of estuaries. However,
it is difficult to know if these samples are representative of the spatial and temporal
variability in carbonate system parameters. While this time-series study cannot conclude
whether our broader sampling efforts in the MAE are representative of the spatial
variability in the estuary, it can investigate how representative our bimonthly to monthly



sampling is of the more high-frequency temporal variability that the Aransas Ship
Channel experiences.

One-way ANOVAs were conducted to compare between monitoring methods

(separate one-way ANOVAs within each season because of the significant interaction
between these factors in an initial two-way ANOVA). There were three levels of
monitoring method included in the comparison of means: continuous monitoring, discrete
monitoring during only the continuous monitoring period, and discrete monitoring over
the entire period (C, $D_C$, and D, respectively, in Table 3). To interpret the results, a
difference in means between the continuous monitoring and discrete monitoring datasets
would only indicate that the 10-month period of continuous monitoring was not
representative of the 5+ year period that discrete samples have been collected, but a
difference in means between the continuous data and discrete sample data collected
during the continuous monitoring period represents discrepancies between types of
monitoring.

There were several instances where seasonal parameter means significantly

differed between the 10-month continuous monitoring period and the 5+ year discrete
sampling period (Table 3, $C \neq D$ or $D_c \neq D$) including temperature in the summer and fall,
salinity in the spring, pH in the summer and fall, and $pCO_2$ in winter, spring, and summer.
While clear seasonal variability was demonstrated for most parameters (using both
continuous and discrete data for the entire period), these differences between the 10-
month continuous monitoring period and our 5+ year monitoring period illustrate that
there is also interannual variability in the system. Therefore, short periods of monitoring
are unable to fully capture current baseline conditions.



During the continuous monitoring period (2016-2017), we found no significant

difference between sampling methods in the seasonal mean temperature, salinity, or
$p$CO$_2$. The two sampling methods also resulted in the same mean pH for all seasons
except for summer, when the sensor data recorded a higher mean pH than discrete
samples (Tables 1 and 3). During this case, we can conclude that discrete monitoring did
not accurately represent the system variability that was able to be captured by the sensor
monitoring. However, given that most seasons did not show differences in pH or $p$CO$_2$
between sampling methods, the descriptive statistics associated with the discrete
monitoring did a fair job of representing system means. This is evidence that long-term
discrete monitoring efforts, which are much more widespread in estuarine systems than
sensor deployments, can be generally representative of the system despite known
temporal variability on shorter time scales.

Understanding the relationships of pH and $p$CO$_2$ with temperature and salinity is

important in a system. Both the continuous and discrete sampling types indicate that pH
has a significant negative relationship with both temperature and salinity and $p$CO$_2$ has a
s significant positive relationship with both temperature and salinity (Fig. 7). Based on
the results of an Analysis of Covariance (ANCOVA), the relationship (slope) of pH with
both temperature and salinity and of $p$CO$_2$ with salinity were not significantly different
between types of monitoring (considering the sensor deployment period only), supporting
the effectiveness of long-term discrete monitoring programs when sensors are unable to
be deployed. However, ANCOVA did reveal the relationship of $p$CO$_2$ with temperature is
significantly different (method:temp p=0.0062) between monitoring methods.



While *in situ* monitoring is usually lacking good spatial coverage, it is effective in
capturing temporal resolution and presumably providing better estimates of average $CO_2$
flux at a given location versus periodic sampling. Previous studies have pointed out that
discrete sampling methods, which generally involve only daytime sampling, do not
adequately capture the diel variability in the carbonate system and may therefore lead to
underestimation of $CO_2$ fluxes. However, we found no significant difference (within any
season) between $CO_2$ flux values calculated with sensor data versus discrete samples
(Table 3). Calculated $CO_2$ fluxes also did not significantly differ between day and night
during any season, despite some differences in $pCO_2$ (Table 2), likely due to the large
error associated with the calculation of $CO_2$ flux (Table 1, Fig. 5) which will be further
discussed below. Therefore, the expected underestimation of $CO_2$ flux based on diel
variability of $pCO_2$ was not encountered at our study site, validating the use of discrete
samples for quantification of $CO_2$ fluxes (until methods with less associated error are
available). Even given less error in calculated flux, estimated fluxes would likely not
differ between methods on an annual scale (as $pCO_2$ did not), but $CO_2$ fluxes may differ
on a seasonal scale since the differences between daytime and nighttime $pCO_2$ were not
consistent across seasons (Table 2).
There are many factors contributing to error associated with $CO_2$ flux. There is
still large error associated with estimates of estuarine $CO_2$ flux because turbulent mixing
is difficult to model and turbulence is the main control on $CO_2$ gas transfer velocity, $k$, in
shallow water environments. Thus, our wind speed parameterization of $k$ is imperfect and
likely the greatest source of error. Other notable sources of error include the data
treatment. For example, we chose to seasonally weight the individual calculated flux





values in the calculation of annual flux to account for differences in sampling frequency
between seasons. From continuous data, the weighted average flux was 0.2 mmol m$^{-2}$ d$^{-1}$,
although choosing not to seasonally weight and simply look at the arithmetic mean of
fluxes calculated directly from sampling dates would have resulted in an annual $CO_2$ flux
of -0.7 mmol m$^{-2}$ d$^{-1}$ for the same period. Similarly, the weighted average flux from all
5+ years of discrete data was -0.9 mmol m$^{-2}$ d$^{-1}$, but the arithmetic mean of fluxes would
have resulted in an annual $CO_2$ flux of 0.2 mmol m$^{-2}$ d$^{-1}$ for the same period. Another
source of error that could be associated with the calculation of flux from the discrete data
is the way in which wind speed data are aggregated to be used in the windspeed
parameterization. We decided to use daily averages of the windspeed for calculations.
Using the windspeed measured for the closest time to our sampling time or the monthly
averaged wind speed may have resulted in very different flux values.
*4.3.2 Direct agreement of measurement methods and quantified uncertainties associated*
*with parameters*

Direct comparisons were made between measurements from sensors and

laboratory-analyzed bottle samples—including both quality control (QC) samples taken
from the cooler that housed the sensors at the time when these sensors took recorded
readings and long-term monitoring samples taken from the ship channel near the sensors
(within 100 m) that occurred at various times and were compared to sensor measurements
of the closest full hour (Table 8). The mean difference between the SeaFET pH
measurements and the QC samples (continuous – discrete) prior to sensor data correction
was 0.05 ± 0.08 (Table 8, which would reduce to 0.00 ± 0.08 following the correction).
The mean difference between the SAMICO2 $pCO_2$ measurements and the QC samples



(continuous – discrete) was -18 ± 44 (Table 8) when discrete sample $pCO_2$ was calculated
using Millero (2010) constants. We used several different constants to calculate $pCO_2$ to
check this offset; all were similar in mean and standard deviation, but the offset could be
slightly reduced using Millero (2002) constants.
Table 8. Comparison of discrete and continuous monitoring. The difference between
sampling methods is reported in two different ways: the difference between sensor
measurements and laboratory measurement of quality control (QC) bottle samples taken
directly from the cooler (here the pH difference is prior to the sensor pH correction of
+0.05), and the difference between sensor measurements and laboratory measurement of
bottle samples taken from a nearby station for our 5+ year monitoring (here the pH
difference if after the sensor pH correction of +0.05, see methods for details). For all
calculated parameters, dissociation constants from Millero 2010 were used. Error—
analytical error for directly measured parameters and propagated error for calculated
parameters (mean ± standard deviation, calculated in the seacarb package in R—
associated with carbonate system variables is also reported.

| | Difference between sampling methods (mean difference ± standard deviation of the difference) | | Error (Analytical or Propagated) | |
|---|---|---|---|---|
| | Sensor – QC cooler samples (prior to sensor pH correction, n=12) | Sensor – discrete samples (after pH sensor correction, n=13) | **Discrete Sampling (n = 104)** | **Continuous Monitoring (n = 6088)** |
| Temperature (°C) | | | 0.1 | 0.1 |
| Salinity | -0.16 ± 1.44 | 0.50 ± 1.69 | 0.01 | 0.1 |
| pH | -0.05 ± 0.08 | 0.01 ± 0.12 | 0.0004 | 0.05 |
| $pCO_2$ (µatm) | -18 ± 44 | 25 ± 63 | 7 ± 2 | 1.0 |
| DIC (µmol kg⁻¹) | | | ±2.5 | 327.4 ± 63.2 |
| TA (µmol kg⁻¹) | | | 7.4 ± 0.9 | 400.7 ± 81.0 |
| $\Omega_{Ar}$ | | | 0.19 ± 0.03 | 1.08 ± 0.31 |


Given that the analytical accuracy of the SeaFET instrument is 0.05 pH units, the
average offset between sensor and laboratory values of quality control samples
demonstrates fair agreement (Table 8). Given that calculated uncertainty associated with
calculated discrete $pCO_2$ was 7 ± 2, we did not see great agreement between SAMICO2
$pCO_2$ and laboratory-calculated $pCO_2$ for quality control samples (mean difference of -18
± 44, Table 8). Mean offsets and their associated standard deviations were larger when





comparing sensor data to samples taken during our long-term discrete monitoring effort.
This is not surprising given that the discrete sample collection did not occur at the exact
time of the sensor measurement or the exact location of the cooler pump inlet. Greater
sensor-laboratory agreement has been achieved for open ocean settings, but this larger
standard deviation is likely a result of the temporal variability in the more complex
estuarine environment where these instruments have been much less widely deployed to
date.

Propagation of error associated with computed carbonate system parameters was

done using the *seacarb* package in R (Gattuso et al., 2018); the error propagation includes
error associated with the measurements of the input pair (1 µatm for $p$CO$_2$ from
SAMICO2 and 0.05 for pH from SeaFET; 0.0004 for laboratory spectrophotometric pH
and 2.5 µmol kg$^{-1}$ for laboratory DIC), error associated with *in-situ* temperature (0.1 °C)
and salinity (0.1 for sensor-measured and 0.01 for laboratory-measured), and error
associated with total boron the key dissociation constants (standard recommended error
used) (Table 8). While the error associated with calculated parameters from discrete
bottle samples was relatively small and likely a result of uncertainties in constants (Orr et
al., 2018), we note that the error associated with calculated dissolved inorganic carbon
(DIC), total alkalinity (TA), and saturation state of aragonite ($\Omega_{Ar}$), which are other
frequently addressed carbonate system parameters, was large when calculated with sensor
data. This large error is likely a result of both the relatively low analytical precision
associated with the pH sensor and the poor mathematical combination of variables for
speciation calculations. Hence, we limited the discussion to pH (which was directly
measured for both continuous monitoring and laboratory analysis of discrete samples)



and $pCO_2$ (which was directly measured for continuous monitoring and had relatively
low error when calculated with discrete sample DIC and pH, Table 8) and omitted any
discussion of the parameters with high propagation error. The high error suggests that it
may be important to develop and broadly use autonomous sensors that can measure
carbonate system parameters that allow for lower propagated error to have a full picture
of estuarine carbonate chemistry on high-frequency time scales.
**5. Conclusions**
We monitored carbonate chemistry parameters (pH and pCO₂) using both sensor
deployments (10 months) and discrete sample collection (5+ years) at the Aransas Ship
Channel, TX, to characterize temporal variability and investigate controlling factors.
Both sampling methods demonstrated significant seasonal variability at the location, with
highest pH (lowest $pCO_2$) in the winter and lowest pH (highest $pCO_2$) in the summer.
Significant diel variability was also evident from sensor data, though diel fluctuations
were smaller than many other areas previously studied. Carbonate chemistry parameters
were among the most important environmental parameters to distinguish between both
diel and seasonal environmental conditions.
The difference between daytime and nighttime values of carbonate system
parameters varied between seasons, occasionally reversing the expected diel variability
due to biological processes. It was evident that biological activity is not the strongest
controlling factor of diel variability at this location, likely surpassed by tidal control
despite the small tidal range in the northwestern GOM. Controls on the system also
differed over different time scales, with temperature becoming a less important control
over shorter time scales.



976   Tides exerted significant control on the carbonate system, and low tide allowed

977 more biological control of the system. Higher mean $p\mathrm{CO_2}$ was reported for low tide

978 versus high tide across all seasons. pH was higher at high tide during winter and summer

979 but deviated from the expected pattern during spring with lower pH during high tide. The

980 results suggest that the controlling factors of the carbonate system may not be exerted

981 equally on both pH and $p\mathrm{CO_2}$ on diel timescales, causing separation of their diel or tidal

982 relationships during certain seasons. The detailed investigation of controlling factors

983 provides strong support for the implementation of carbonate chemistry monitoring in

984 conjunction with preexisting coastal environmental monitoring infrastructure, which has

985 had little application in estuarine environments thus far.

986   Despite known temporal variability on shorter timescales, discrete sampling was

987 generally representative of the average carbonate system on a seasonal and annual basis

988 based on comparison with our sensor data. Additionally, there was no difference in $\mathrm{CO_2}$

989 flux between sampling types supporting the validity of discrete sample collection for

990 carbonate system characterization.

991 **Data availability**

992 Continuous sensor data are archived with the National Oceanic and Atmospheric

993 Administration's (NOAA's) National Centers for Environmental Information (NCEI)

994 (https://doi.org/10.25921/dkg3-1989). Discrete sample data are available in two separate

995 datasets archived with National Science Foundation's Biological & Chemical

996 Oceanography Data Management Office (BCO-DMO) (doi:10.1575/1912/bco-

997 dmo.784673.1 and doi: 10.26008/1912/bco-dmo.835227.1).



**Author Contribution**
MM and XH defined the scope of this work. XH received funding for all components of
the work. MM, HY, and CJS performed field sampling and laboratory analysis of
samples. MM prepared the initial manuscript and all co-authors contributed to revisions.
**Competing interests**
The authors declare that they have no conflict of interest.

**Acknowledgements**
Funding for autonomous sensors and sensor deployment was provided by the
United States Environmental Protection Agency's National Estuary Program via the
Coastal Bend Bays and Estuaries Program Contract No. 1605 Thanks to Rae Mooney
from Coastal Bend Bays and Estuaries Program for assistance in the initial sensor setup.
Funding for discrete sampling as well MM's dissertation research has been supported by
both NOAA National Center for Coastal Ocean Science (Contract No.
NA15NOS4780185) and NSF Chemical Oceanography Program (OCE-1654232). We
also appreciate the support from the Mission-Aransas National Estuarine Research
Reserve in allowing us the boat-of-opportunity for our ongoing discrete sample
collections and the University of Texas Marine Science Institute for allowing us access to
their research pier for the sensor deployment. A special thanks to Hongjie Wang, Lisette
Alcocer, Allen Dees, and Karen Alvarado for assistance with field work.

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
