# Peer review of "Estuaries are complex systems with substantial heterogeneity in water chemistry, including carbonate chemistry parameters such as pH and partial pressure of CO2 (pCO2), because of the diversity of co-occurring biogeochemical processes. To better understand estuarine acidificati"

_Biogeosciences, 2021_

## Referee Comment (RC1)

**Overarching thoughts:**

In general, the work is an important contribution to a developing field of understanding biogeochemical variability in coastal zones and in particular, estuaries. The authors have a robust, long-term data set that merits publishing. I think much of the information could be trimmed down to focus primarily on the carbonate chemistry and air-sea fluxes, and a lot of information in tables could be moved to supplementary material to streamline this and keep the more novel aspects of the work at the forefront. This would also aid with the overall length. Additionally, see cautionary comments on the air-sea flux estimates. Authors could also benefit from some re-organization within the methods discussion to clearly define all data streams being used, and a separate section for applied statistical methods. Much of this information is buried in the discussion.

**Intro:**

49-50: Could specify that *an increase* or *addition of* CO2 acidifies seawater.

56-57: Vague, I would remove or specify. As it reads now, you say that 'the speciation of inorganic carbon in seawater is important […] because the ocean contributes substantially to the global carbon budget, which is important to understand due to climate change". What do you mean? I see the motivation behind understanding C chem variability but not so sure about the carbon budget point here.

70: Maybe I am missing something here: 'mean diel ranges exceed 0.1 but single day ranges can exceed 1' – isn't the 0.1 value stated here somewhat meaningless if the time frame over which you are averaging isn't stated? (Can you add "mean *seasonal* *weekly* or *monthly* ranges can exceed 0.1 and single day ranges can exceed 1 pH unit" or something along these lines).

93: What are "relatively high CO2 fluxes" (do you mean *influx, efflux*, or just generally larger magnitude fluxes regardless of direction?).

**Materials and Methods:**

Figure 1: Can you add a scale bar. From the yellow arrow it looks like the study actually did not take place in the estuary, but rather just outside. It is interesting because the paper to this point feels framed around estuarine C chem, but the location of the sensors based on Fig. 1 is an in-between spot, that seems more in the GOM than the estuary itself. This is fine, but the discussion to follow will need to carefully discuss GOM versus estuarine influences as this clearly will get a lot of both.

*Section 2.2:* Could provide a few more details on calibration procedures in this section – see Rivest et al. (2016) or Bresnahan et al. (2014). Was the -0.05 correction applied across the entire dataset as a calibration, or was this a slow drift? If the latter, how was this correction performed?

*Section 2.4*
The air-sea flux methods make a lot of assumptions. Recognizing that these can be difficult to measure, I think the overarching methods applied are acceptable, but should be some careful discussion or sensitivity analysis performed. For example, alterations in the air-sea gradient can drastically alter the estimated fluxes. I am not sure that using global average atmospheric xCO2 values is appropriate, but I have not run the sensitivity analyses to check. How often were these global values applied when local data were not available? If infrequently, it would be good to state and authors should discuss the continuity between these values. For instance, if the mean diel range in seawater pCO2 was 58 uatm, and the variability in the global estimates from the air-side concentrations eclipse this seawater variability, this would not be appropriate. Similarly, reporting the distance between the sampling site and the stations that wind and CO2 data were pulled from can help readers ID how large these assumptions are. Can authors clearly state over what frame data are averaged as well? Are you using hourly avg wind and CO2 data in calculations? Daily? Etc.

Standard parametric statistical test applied here, no time series analysis or more advanced techniques – may be worth noting that all data met assumptions and if not were there any transformations etc.

A very brief background on the T/B methods might be helpful as many of this paper's discussion rests on this

~145: What is the sampling duration and frequency? As in (e.g.) is each 1 hr measurement had a single collection point, once per minute then averaged over the entire hour?

~162: Were discrete/bottle samples for pH and pCO2 also collected from the cooler and taken back to the lab for these analyses or for comparison with the sensors' continuous sampling? If so, where are these methods? (you go on in the next section to describe discrete sampling, but it is unclear if these discrete sampling data were used in any sensor calibration procedures? Or just to corroborate uncorrected sensor data?) It is unclear if these discrete samples are their own analysis or not.

258: Takahashi et al. 2002 is not in the works cited.

**Results:**

There is a lot of information in tables, some of which might be better placed into a supplementary table to reduce length.

Figure 2: Figure text states that panels E-H covers the same dates as panels A-D, but the x axes do not match this (typo in caption).

Table 1: Why do the winter CO2 flux values have one value in parentheses only? I am confused by the caption "CO2 fluxes were calculated using the Jiang et al. (2008) wind speed

parameterization for gas transfer velocity, and ranges of CO2 flux that are given in brackets represent means calculated using parameterizations from Ho et al. (2006) and Raymond and Cole (2001), respectively." What do you mean by respectively? Do you mean that Ho et al. are in brackets and Raymond and Cole are in parentheses? Or why are some values in both parentheses and brackets? Not getting the coding here.

Table 2: You note significant difference by the p-value that summer temperature are higher at night than during the day. While true according to your statistical test, the error bar will overlap very considerably, I would check you test and if correct, make statements accordingly – the difference in reality seems marginal.

Table 3: Figure caption describes a lot of the statistical methods. I would remove from the figure captions and place into the statistical analyses section above (I might suggest making statistical analyses its own section entirely, rather than mixing this information in the same section as the air-sea fluxes etc). For example, just define alpha in this section, you already stated you will use a Westfall adjustment, and if described in the methods, you do not need to describe this or why individual one-way ANOVAs were conducted in the caption.

Table 5: Are these mean +/- SE or SD? Can authors provide the sample size in addition to the p-value for the t-tests run? Hard to judge based on p values alone. Additional test statics and pertinent info could be placed into a supplementary table.

447: Format error (one extra tab)

**Discussion:**

508-509: Does this really remove the influence of tides? It seems more like you can just analyze controls during these tidal periods rather than you removing their influences? Rephrasing might fix this.

538-551: This is the first mention of chlorophyll and these other env. parameters being used for further analysis. This should be in the methods, in addition to the statistical and data processing techniques used in analysis (ie Pearson's correlation in Table 6).

560-561: Why is water temperature warmer at night? Isn't this unusual? Can you explain why?

579: Spelling (asterisks)

605-611: To my earlier point about air-sea flux methods, I wonder if these results would change with more precise estimates of atmospheric pCO2 levels.

632-635: Given the analysis, it seems like rather than running isolated linear regression between two factors, the authors could consider a 'full' model where available data across all the aforementioned parameters are included and those that are significant are selected. This would be a more typical approach to explain variation.

651: Typo ('both pCO2 and pH')

657: Belongs in methods

684-685: What do you mean by tidal control? As in, in the winter, the nighttime typically corresponded with tidal phases that carry lower pCO2 water into the sampling site at night? Some discussion of mechanism might be worthwhile here.

700: As mentioned above, if DO is part of this analysis, needs to be mentioned and described before this point. Line 706 feels like it belongs in the results.

719-727: I feel this is beyond the scope of this paper and detracts from the main point. If authors insist on including, these methods also need to be described earlier. It is possible this could be move to supplementary information, but I do not think this is necessary. Also – do you think that if it is an urban water way, this boat traffic would further elevate atmospheric CO2 levels? Would this be captured by the station you pull data from?

728-738: This feels like it belongs on conclusions or could be removed altogether. At present, it seems misplaced.

743-747: Methods/statistical analyses. (also – I see you got to the 'full' model approach I suggested above eventually. If the methods are more clearly described, then readers will knows this up front.).

796: Can the authors not find any references other than pers comms to compare their variability to? It seems like this statement could be backed up with references as well.

821-824: Extraneous. This has already been described.

862: When you say *in situ* monitoring, do you mean continuous, sensor-based sampling versus periodic sampling?

870: Yes – I agree with authors on this point. If other methods such as eddy covariance were employed to estimate air-sea fluxes, you may find diurnal differences (I have, also in an estuary). But of course, maybe not! Not sure of this system. Generally, I appreciate the discussion of the sources of error on these measurements that follows.

937-958: This seems a bit in the weeds and not very relevant to this MS. Can this be removed or streamlined.

1003: Font typo

1008: Missing period after contract No. 1605.

---

## Referee Comment (RC2)

This manuscript describes and analyzes a study of variability in and drivers of carbonate chemistry parameters in the Gulf of Mexico. The high temporal resolution data sets used in this study are very valuable to the field of coastal carbonate chemistry, both individually as long-term discrete and short-term continuous data sets, and when compared against one another. The authors do a nice job of analyzing the seasonal and interannual relationships between the carbonate system and its drivers, as well as critically analyzing the data they use. The current organization and length of the manuscript detract from its message, but it will be a strong contribution after revision. Please see below for specific comments.

Line 56-60: these two sentences are repetitive and could be combined into one thought.

Line 110-111: How does this tidal range compare to other estuaries?

Line 292-294: the current setup of these reported values is slightly confusing at first glance. My eye saw them as negatives even though the dash is not connected to the values. Consider altering the style to avoid confusion (e.g., salinity: 30.8 +- 3.7 instead of salinity - 30.8 +-3.7).

Table 1: The column labels here don't line up with the data columns. Perhaps add a vertical line between the leftmost and center data columns to indicate that center and right columns are both discrete sampling. Or, re-align the column headings.

Table 1, 2, 3, 4, 5, 6: Consider moving these to a supplemental. You can still report important values in the text but having all the data displayed in large tables is clunky.

Line 354: Why is temperature higher at night?

Line 390-397: The T/B values varied over different time scales, but the overall pattern did not. It might be more useful to point out succinctly that non-thermal processes had more control than thermal processes over the entire sampling period, in each season individually, and on most days (11/178).

Line 419: change "season" to "seasons"

Lines 390-397, 452-459:Why did continuous vs. discrete data show different T/B patterns?

Line 482-483: This paragraph could be re-organized. The first two sentences describe the small relative difference between sampling methods over the 10-month period. The rest of the paragraph describes sampling bias between methods. I'd recommend starting with "The difference in T/B between sampling methods was small over a 10-month period, but sampling methods did not align over shorter, seasonal time scales." Then talk about the reasons for bias.

Section 4.1.2: The linear regressions described in this section don't necessarily aid in describing the system as a whole. They are interesting to see but are overshadowed by the linear discriminant analysis described in section 4.2. I suggest removing this section or cutting it down to its most pertinent pieces. Specific recommendations below:

Much of the text in this section should be moved to methods (lines 536-551, 598-600, 657-659, 719-723,) or results (lines 606-608, 642-649, 664-671, 698-701, 705-706, 723-727).

Line 548: Did omitting stratified water bias observations toward any particular season? I'd assume stratification is strongest in summer and therefore more summer observations would be omitted, but perhaps that is not the case in the Gulf.

Line 657: indent if this is a new paragraph

Lines 728-738: This paragraph should go in the conclusions section – it is a strong summary of the strength of this work

Section 4.2: I think the analysis and results you describe in this section is the most impactful in the study. The linear discriminant (LD) analysis needs to be described in the methods (lines 743-747, 750-755). Table 7 and related text should go in the results.

Section 4.3: This comparison between monitoring methods is valuable, especially given the conclusion that long-term discrete monitoring is generally representative of the system (lines 848-851). However, the authors have already discussed some of the information in this section, including the sampling methods and some results of the sampling method comparison (in section 4.1.1 when discussing differences in reported T/B values). I suggest either moving this section to the beginning of the discussion to set the stage for your discussion of data collected using the different methods, or moving the T/B sampling method discussion points to this section.

Lines 937-958: This paragraph could be pared down or removed. It is nice to understand the thinking behind omitting other carbonate system parameters but since they are not mentioned previously, it almost seems extraneous.

Conclusion: I recommend that the focus should be shifted to the significance of this study, rather than a summary of parameter-by-parameter results. The paragraph in lines 728-738 would fit in nicely. Lines 982-988 are strong conclusions that show the significance of the study as well.

---

## Author Response (AR1)

**We thank both reviewers for their constructive feedback.**

**Based on feedback from both reviewers, we substantially edited the organization of the manuscript - moving much of the text from the discussion into the methods and results sections. We also changed the results section breakdown (from the separation of continuous and discrete data results) to subsections related to seasonal variability, diel variability, and controlling factors and correlates (each incorporating both continuous and discrete data results). We believe that this is a much more streamlined arrangement with better flow and less redundancy, significantly improving the paper.**

**Replies to each comment by reviewers are listed below:**

*REVIEWER #1*

*Overarching thoughts:*
*In general, the work is an important contribution to a developing field of understanding biogeochemical variability in coastal zones and in particular, estuaries. The authors have a robust, long-term data set that merits publishing. I think much of the information could be trimmed down to focus primarily on the carbonate chemistry and air-sea fluxes, and a lot of information in tables could be moved to supplementary material to streamline this and keep the more novel aspects of the work at the forefront. This would also aid with the overall length. Additionally, see cautionary comments on the air-sea flux estimates. Authors could also benefit from some re-organization within the methods discussion to clearly define all data streams being used, and a separate section for applied statistical methods. Much of this information is buried in the discussion.*

**Response:**
We included information about all data used and all statistical analyses used in the methods section of the paper. Several tables and figures were moved to the supplementary materials to help reduce the length of the paper, and we also trimmed a lot of the discussion to keep the focus to the most important aspects.

*Intro:*
*49-50: Could specify that \*an increase\* or \*addition of\* CO2 acidifies seawater.*

**Response:**
This has been changed the include "*an addition of*". Now line 47.

*56-57: Vague, I would remove or specify. As it reads now, you say that 'the speciation of inorganic carbon in seawater is important [...] because the ocean contributes substantially to the global carbon budget, which is important to understand due to climate change". What do you mean? I see the motivation behind understanding C chem variability but not so sure about the carbon budget point here.*

**Response:**

It was intended to mention the importance of marine waters as a sink/source of $CO_2$ to the atmosphere, given the effects of changes to atmospheric $CO_2$ concentrations, but we have decided to remove this sentence.

*70: Maybe I am missing something here: 'mean diel ranges exceed 0.1 but single day ranges can exceed 1' – isn't the 0.1 value stated here somewhat meaningless if the time frame over which you are averaging isn't stated? (Can you add "mean \*seasonal\* \*weekly\* or \*monthly\* ranges can exceed 0.1 and single day ranges can exceed 1 pH unit" or something along these lines).*

**Response:**
This sentence has been edited to say "*Mean diel ranges in pH can exceed 0.1 unit in many coastal environments, and especially high diel ranges (even exceeding 1 pH unit) have been reported in biologically productive areas or areas with higher mean $pCO_2$.*" Now lines 62 – 64.

*93: What are "relatively high CO2 fluxes" (do you mean influx, efflux, or just generally larger magnitude fluxes regardless of direction?).*

**Response:**
This has been edited to efflux. Now line 84.

*Materials and Methods:*
*Figure 1: Can you add a scale bar. From the yellow arrow it looks like the study actually did not take place in the estuary, but rather just outside. It is interesting because the paper to this point feels framed around estuarine C chem, but the location of the sensors based on Fig. 1 is an in between spot, that seems more in the GOM than the estuary itself. This is fine, but the discussion to follow will need to carefully discuss GOM versus estuarine influences as this clearly will get a lot of both.*

**Response:**
The map has been updated to include a scale bar as well as additional relevant information.

The location is within the ship channel, so it is a tidal inlet that does receive influence from both estuarine waters and GOM waters based on the tides. We have slightly altered the introduction text to reflect the influence of coastal waters.

*Section 2.2: Could provide a few more details on calibration procedures in this section – see Rivest et al. (2016) or Bresnahan et al. (2014). Was the -0.05 correction applied across the entire dataset as a calibration, or was this a slow drift? If the latter, how was this correction performed?*

**Response:**
The -0.05 correction was applied across the entire dataset. This was what was suggested as best practice in Bresnahan et al. (2014) when they recommended applying a single calibration point. (The citation has now been added to the text). To confirm that this was the best approach, we showed that there was not a trend in the direction or magnitude of offset between the deployed SeaFet sensor and our laboratory measurements of pH over the 10-month monitoring period.

This has been better explained in the text. Now lines 124 – 128 and supplemental materials lines 81 – 96.

*The air-sea flux methods make a lot of assumptions. Recognizing that these can be difficult to measure, I think the overarching methods applied are acceptable, but should be some careful discussion or sensitivity analysis performed. For example, alterations in the air-sea gradient can drastically alter the estimated fluxes. I am not sure that using global average atmospheric xCO2 values is appropriate, but I have not run the sensitivity analyses to check. How often were these global values applied when local data were not available? If infrequently, it would be good to state and authors should discuss the continuity between these values. For instance, if the mean diel range in seawater pCO2 was 58 uatm, and the variability in the global estimates from the airside concentrations eclipse this seawater variability, this would not be appropriate. Similarly, reporting the distance between the sampling site and the stations that wind and CO2 data were pulled from can help readers ID how large these assumptions are. Can authors clearly state over what frame data are averaged as well? Are you using hourly avg wind and CO2 data in calculations? Daily? Etc.*

**Response:**
We understand the reviewer's concern and need for additional clarification to fully understand the assumptions that were made.

We have more clearly stated the way in which data were averaged for flux calculations. Lines 189 – 193 (wind speed) and Lines 198 – 199 ($xCO_2$).

We moved some of the information that had been in the $CO_2$ flux calculation methods to the supplementary materials to help streamline the manuscript, and we also expanded/clarified the methods in the supplementary materials to make all assumptions clear and justify our methods (Supplemental materials section: ***Additional information on $CO_2$ flux calculations and windspeed data***; Lines 148 – 171)

*Standard parametric statistical test applied here, no time series analysis or more advanced techniques – may be worth noting that all data met assumptions and if not were there any transformations etc.*

**Response:**
Some, but not all, of the parameters for which we tested means met ANOVA's assumptions of normality and homoscedasticity. Since there are many different comparisons of means that were made, we wanted to streamline by using the same statistical tests for all comparisons rather than using a non-parametric alternative for certain tests. Therefore, for those that did not meet the assumptions, we relied on the fact that ANOVA is fairly robust to the violation of assumptions. Transformations were not noted, as data transformations did not fully fix violations and were not used.

*A very brief background on the T/B methods might be helpful as many of this paper's discussion rests on this*

**Response:**
The following sentence was added to provide additional background. "*Over any given time period, this method uses the ratio of the ranges of temperature-normalized pCO₂ (pCO₂,ₙₜ, Eq. 2) and the mean annual pCO₂ perturbed by the difference between mean and observed temperature (pCO₂, ₜ, Eq. 3) to calculate the relative influence of non-thermal and thermal effects on pCO₂ (T/B, Eq. 4).*" Now lines 221 – 225.

*~145: What is the sampling duration and frequency? As in (e.g.) is each 1 hr measurement had a single collection point, once per minute then averaged over the entire hour?*

**Response:**
We have added the following text to clarify the sensor measurements "*All hourly data were single measurements taken on the hour.*" Now lines 123 - 124

*~162: Were discrete/bottle samples for pH and pCO2 also collected from the cooler and taken back to the lab for these analyses or for comparison with the sensors' continuous sampling? If so, where are these methods? (you go on in the next section to describe discrete sampling, but it is unclear if these discrete sampling data were used in any sensor calibration procedures? Or just to corroborate uncorrected sensor data?) It is unclear if these discrete samples are their own analysis or not.*

**Response:**
We now discuss this in more detail in the supplementary materials. We have added the text "*The same methods used for discrete sampling analysis apply for these samples (see Section 2.3)*" to clarify how these quality control discrete samples were analyzed. The long-term discrete sampling data were not used for the direct QA/QC since they were not always necessarily taken on the top of the hour and did not include samples pulled from the cooler where sensors were housed. See supplemental materials section ***Sensor data correction and direct agreement of measurement methods*** (Lines 80 – 86)

*258: Takahashi et al. 2002 is not in the works cited.*

**Response:**
Thank you for catching this mistake. This citation has now been added.

*Results:*
*There is a lot of information in tables, some of which might be better placed into a supplementary table to reduce length.*

**Response:**
We agree that the number of tables was excessive for the body of the manuscript, especially given that most of the most pertinent information can be gleaned from the figures. We have moved many of the tables to supplemental materials.

*Figure 2: Figure text states that panels E-H covers the same dates as panels A-D, but the x axes do not match this (typo in caption).*

**Response:**
Thank you for catching that. It has been fixed. (Now Fig. S1)

*Table 1: Why do the winter $CO_2$ flux values have one value in parentheses only? I am confused by the caption "$CO_2$ fluxes were calculated using the Jiang et al. (2008) wind speed parameterization for gas transfer velocity, and ranges of $CO_2$ flux that are given in brackets represent means calculated using parameterizations from Ho et al. (2006) and Raymond and Cole (2001), respectively." What do you mean by respectively? Do you mean that Ho et al. are in brackets and Raymond and Cole are in parentheses? Or why are some values in both parentheses and brackets? Not getting the coding here.*

**Response:**
All negative flux values were given in parentheses to more clearly separate the dashes from the negative signs. However, since the Raymond and Cole (2001) estimates were not exclusively larger than the Ho et al. (2006) estimates, we decided to remove the dash that indicated a range and just give the two values in brackets; this eliminated the need to have parentheses throughout, which makes the whole flux section of the table appear a little cleaner. The table caption was also altered for clarity and to reflect changes.  (Now Table S1)

*Table 2: You note significant difference by the p-value that summer temperature are higher at night than during the day. While true according to your statistical test, the error bar will overlap very considerably, I would check you test and if correct, make statements accordingly – the difference in reality seems marginal.*

**Response:**
You are correct that there is overlap in the error bars. It is important to note that the statistical test was a paired test (it was comparing the nighttime value to the daytime value on the same day), which better detects a diel relationship than just visually comparing the mean and sd as they are listed in Table S3. The loess figure (Fig 5A) provides a good visualization - points plotted below zero were those individual days when temperature was greater during the nighttime. You can see that this was usually (but not always) the case during the seasons that did show a significant difference in Table S3 (summer and fall). For example, there were 74 days sampled in the summer. Of those, 51 days had a nighttime temperature that was greater than daytime (greater by $0.50 \pm 0.36$), while 23 days had the nighttime temperature that was less than daytime (less by $0.20 \pm 0.19$).

We have changed the text to acknowledge that the significant difference does not indicate that nighttime temperature is higher across the board. "*We note that significant differences in day and night temperature within seasons do not indicate that diel differences were observed on all days within the season, as large standard deviations in both daytime and nighttime values result in considerable overlap*". Now lines 588 – 591.

*Table 3: Figure caption describes a lot of the statistical methods. I would remove from the figure*

*captions and place into the statistical analyses section above (I might suggest making statistical analyses its own section entirely, rather than mixing this information in the same section as the air-sea fluxes etc). For example, just define alpha in this section, you already stated you will use a Westfall adjustment, and if described in the methods, you do not need to describe this or why individual one-way ANOVAs were conducted in the caption.*

**Response:**
We appreciate this feedback. We have added a statistical analysis section to the methods (section 2.6, lines 264 – 317), and we have removed all repetitive information from table and figure captions.

*Table 5: Are these mean +/- SE or SD? Can authors provide the sample size in addition to the pvalue for the t-tests run? Hard to judge based on p values alone. Additional test statics and pertinent info could be placed into a supplementary table.*

**Response:**
This is mean ± standard deviation (now denoted in the caption). We also removed additional repetitive information about the statistics from the caption. Sample size was the same for each parameter within each tide level/season combination, so rather than adding a repetitive column to the table, sample size values have been denoted in the statistical analysis section of the methods. We think this provides ample information for interpretation and did not include any additional values in the supplemental materials.

*447: Format error (one extra tab)*

**Response:**
Thank you for pointing out this mistake. It has been corrected.

*Discussion:*
*508-509: Does this really remove the influence of tides? It seems more like you can just analyze controls during these tidal periods rather than you removing their influences? Rephrasing might fix this.*

**Response:**
We agree with you that saying that this removes the influence of tides is not the best way to phrase this. It has been reworded to say "*We calculated the thermal and non-thermal $pCO_2$ terms separately during high tide and low tide periods and found that non-thermal control is more important during low tide conditions (within each season T/B is $0.10 \pm 0.07$ lower during the low tide than high tide)*" Now lines 620 – 623.

*538-551: This is the first mention of chlorophyll and these other env. parameters being used for further analysis. This should be in the methods, in addition to the statistical and data processing techniques used in analysis (ie Pearson's correlation in Table 6).*

**Response:**
Additional information has been added to the methods section

*560-561: Why is water temperature warmer at night? Isn't this unusual? Can you explain why?*

**Response:**
This is also a pattern that we did not expect to see, and we do not have a clear driving explanation for this.

It is possible that there is thermal stratification of the surface waters that occurs during the day and is broken down at night that was not captured since our sensors were measuring ~1 m beneath the surface.

**Response:**
Fixed.

*605-611: To my earlier point about air-sea flux methods, I wonder if these results would change with more precise estimates of atmospheric pCO2 levels.*

**Response:**
Atmospheric $CO_2$ levels generally do not vary greatly over space except for small pockets influenced by metropolitan or industrial areas. We believe that this is of little concern in the area because predominant winds are southeast (coming from offshore), and we also demonstrated that the offset is minimal between the Florida and global datasets that we used.

*632-635: Given the analysis, it seems like rather than running isolated linear regression between two factors, the authors could consider a 'full' model where available data across all the aforementioned parameters are included and those that are significant are selected. This would be a more typical approach to explain variation.*

**Response:**
Thank you for the comment. As you saw later, we did use LDA to explain variation in the system. This text was more a caveat to support that the variations in salinity were not really driven by tide and therefore would primarily indicate freshwater inflow and evaporation/precipitation. The text has been changed to make the intent of the sentence clearer "*Fluctuating salinity at ASC may also result from direct precipitation, stratification, and tidal fluctuations; however, the low $R^2$ (0.02) associated with a simple linear regression between tide level and salinity (p<0.0001) indicates that salinity fluctuations are more indicative of non-tidal factors*" Now lines 658 – 661.

**Response:**
Fixed.

**Response:**
This explanation of loess models has been moved to the methods. Now lines 269 – 271.

*684-685: What do you mean by tidal control? As in, in the winter, the nighttime typically corresponded with tidal phases that carry lower pCO2 water into the sampling site at night? Some discussion of mechanism might be worthwhile here.*

**Response:**
We added text to further explain this mechanism "*i.e., the higher nighttime tides in winter brought in enough low $CO_2$ water from offshore to fully offset any nighttime buildup of $CO_2$ from the lack of photosynthesis.*" Lines 638 – 640.

*700: As mentioned above, if DO is part of this analysis, needs to be mentioned and described before this point. Line 706 feels like it belongs in the results.*

**Response:**
Line 706 has been moved to the results (and reworded to "*No significant difference in daytime and nighttime DO were observed during any season (Fig. 5F; paired t-tests, winter p = 0.1573, spring p = 0.4877, summer p = 0.794)*"). Lines 385-386.

Information about analyses involving additional environmental parameters, including DO, has been moved to the methods. (Section *2.5 Additional data retrieval and data processing to investigate carbonate system variability and controls*)

*719-727: I feel this is beyond the scope of this paper and detracts from the main point. If authors insist on including, these methods also need to be described earlier. It is possible this could be move to supplementary information, but I do not think this is necessary. Also – do you think that if it is an urban water way, this boat traffic would further elevate atmospheric CO2 levels? Would this be captured by the station you pull data from?*

**Response:**
We agree that this is not necessary to include, and we have removed the paragraph.

*728-738: This feels like it belongs on conclusions or could be removed altogether. At present, it seems misplaced.*

**Response:**
We moved this text (about co-locating environmental sensors) to the conclusions.

*743-747: Methods/statistical analyses. (also – I see you got to the 'full' model approach I suggested above eventually. If the methods are more clearly described, then readers will knows this up front.).*

**Response:**
A description of the LDA has been included in the methods section. Now lines 301 – 317.

*796: Can the authors not find any references other than pers comms to compare their variability to? It seems like this statement could be backed up with references as well.*

**Response:**
We removed the information about the other 6 estuaries that were undergoing monitoring at the same time (all part of the EPA/NEP OA monitoring effort) and added citations of other published works. Now lines 548 – 550.

*821-824: Extraneous. This has already been described.*

**Response:**
Agree. Unnecessary lines of text were removed.

*862: When you say in situ monitoring, do you mean continuous, sensor-based sampling versus periodic sampling?*

**Response:**
Yes, you are correct that is what we meant. We have made sure to refer to sampling methods as either continuous or discrete.

*870: Yes – I agree with authors on this point. If other methods such as eddy covariance were employed to estimate air-sea fluxes, you may find diurnal differences (I have, also in an estuary). But of course, maybe not! Not sure of this system. Generally, I appreciate the discussion of the sources of error on these measurements that follows.*

**Response:**
Thank you for your feedback on this topic.

*937-958: This seems a bit in the weeds and not very relevant to this MS. Can this be removed or streamlined.*

**Response:**
While distant from the rest of the manuscript, we believe this discussion of propagated error to be a useful contribution to the literature, as we have not seen it written elsewhere. We decided to move this to the supplemental materials, lines 129 – 142.

*1003: Font typo*

**Response:**
Fixed.

*1008: Missing period after contract No. 1605.*

**Response:**
Fixed.

*This manuscript describes and analyzes a study of variability in and drivers of carbonate chemistry parameters in the Gulf of Mexico. The high temporal resolution data sets used in this study are very valuable to the field of coastal carbonate chemistry, both individually as long-term discrete and short-term continuous data sets, and when compared against one another. The authors do a nice job of analyzing the seasonal and interannual relationships between the carbonate system and its drivers, as well as critically analyzing the data they use. The current organization and length of the manuscript detract from its message, but it will be a strong contribution after revision. Please see below for specific comments.*

**Response:**
Thank you for your thoughtful review.

*Line 56-60: these two sentences are repetitive and could be combined into one thought.*

**Response:**
We decided to remove one of these sentences to streamline the text.

*Line 110-111: How does this tidal range compare to other estuaries?*

**Response:**
The tidal range is smaller than most other estuaries and coastal areas. We changed this sentence to "*The region is microtidal, with a small tidal range relative to many other estuaries, ranging from ~ 0.6 m tides on the open coast to less than 0.3 m in upper estuaries*". Now lines 101 – 103.

*Line 292-294: the current setup of these reported values is slightly confusing at first glance. My eye saw them as negatives even though the dash is not connected to the values. Consider altering the style to avoid confusion (e.g., salinity: 30.8 +- 3.7 instead of salinity - 30.8 +-3.7).*

**Response:**
We agree that it would be an easier read if dashes were converted to colons here, however, we decided to remove this reporting in the revision of the manuscript.

*Table 1: The column labels here don't line up with the data columns. Perhaps add a vertical line between the leftmost and center data columns to indicate that center and right columns are both discrete sampling. Or, re-align the column headings.*

**Response:**
Thank you for this suggestion. We have added some vertical lines to the table to clarify the labels.

*Table 1, 2, 3, 4, 5, 6: Consider moving these to a supplemental. You can still report important values in the text but having all the data displayed in large tables is clunky.*

**Response:**

We agree that the number of tables was excessive for the body of the manuscript, especially given that most of the most pertinent information can be gleaned from the figures. We have moved many of the tables (original tables 1, 2, 3, 4, and 6) to supplemental materials. We believe that table 5 (now Table 2) should remain in the main body of the text.

*Line 354: Why is temperature higher at night?*

**Response:** As replied to the other reviewer: It is possible that there is thermal stratification of the surface waters that occurs during the day and is broken down at night that was not captured since our sensors were measuring ~1 m beneath the surface.

*Line 390-397: The T/B values varied over different time scales, but the overall pattern did not. It might be more useful to point out succinctly that non-thermal processes had more control than thermal processes over the entire sampling period, in each season individually, and on most days (11/178).*

**Response:**
We agree that this text could be greatly reduced. It now reads "*Based on continuous data, non-thermal processes generally exerted more control than thermal processes (T/B<1) over the entire 5+ years of discrete monitoring, within each season, and over most (167/178) days*" Now lines 416 – 419.

*Line 419: change "season" to "seasons"*

**Response:**
Fixed.

*Lines 390-397, 452-459: Why did continuous vs. discrete data show different T/B patterns?*

**Response:**
Primarily, this is because the continuous data capture all extremes while discrete data do not. We added the text "*Continuous monitoring demonstrated a greater magnitude of fluctuation resulting from both temperature and non-thermal processes (i.e., greater $\Delta pCO_{2,t}$ and $\Delta pCO_{2,nt}$), indicating that the extremes are generally not captured by the discrete, daytime sampling, and sensor data would provide a better understanding of system controls.*" Lines 573 – 576.

*Line 482-483: This paragraph could be re-organized. The first two sentences describe the small relative difference between sampling methods over the 10-month period. The rest of the paragraph describes sampling bias between methods. I'd recommend starting with "The difference in T/B between sampling methods was small over a 10-month period, but sampling methods did not align over shorter, seasonal time scales." Then talk about the reasons for bias.*

**Response:**
This was substantially reworked as the entire discussion was reorganized and the discussion of T/B was reduced considerably. Now lines 559 – 597.

*Section 4.1.2: The linear regressions described in this section don't necessarily aid in describing the system as a whole. They are interesting to see but are overshadowed by the linear discriminant analysis described in section 4.2. I suggest removing this section or cutting it down to its most pertinent pieces. Specific recommendations below:*

**Response:**
We now include much less discussion of individual linear regressions.

*Much of the text in this section should be moved to methods (lines 536-551, 598-600, 657-659, 719-723,) or results (lines 606-608, 642-649, 664-671, 698-701, 705-706, 723-727).*

**Response:**
All suggested text to be moved to the methods and results sections have been moved, with the exception of lines 719-723 and lines 723 - 727 (discussion of acid deposition from boat traffic), which was removed based on the recommendation of Reviewer #1.

*Line 548: Did omitting stratified water bias observations toward any particular season? I'd assume stratification is strongest in summer and therefore more summer observations would be omitted, but perhaps that is not the case in the Gulf.*

**Response:**
It disproportionally resulted in the removal of winter observations. Text has been altered to say "*Omitting stratified water reduced our continuous dataset from 6088 to 5524 observations (removing 260 winter, 133 spring, 51 summer, and 120 fall observations), and omitting observations where there were no MANERR data to determine stratification further reduced the dataset to 4112 observations.*" Lines 258 – 261.

*Line 657: indent if this is a new paragraph*

**Response:**
This text has been moved to the methods section.

*Lines 728-738: This paragraph should go in the conclusions section – it is a strong summary of the strength of this work*

**Response:**
This text has been moved to the conclusions section as suggested.

*Section 4.2: I think the analysis and results you describe in this section is the most impactful in the study. The linear discriminant (LD) analysis needs to be described in the methods (lines 743-747, 750-755). Table 7 and related text should go in the results.*

**Response:**

The LDA is now described in the methods section (Now lines 301 - 317) and Table 7 (now Table 1) has been moved to the results. We believe that this positioning of the LDA in a more prominent place in the manuscript does better highlight the impact of this investigation.

*Section 4.3: This comparison between monitoring methods is valuable, especially given the conclusion that long-term discrete monitoring is generally representative of the system (lines 848-851). However, the authors have already discussed some of the information in this section, including the sampling methods and some results of the sampling method comparison (in section 4.1.1 when discussing differences in reported T/B values). I suggest either moving this section to the beginning of the discussion to set the stage for your discussion of data collected using the different methods, or moving the T/B sampling method discussion points to this section.*

**Response:**
Thank you for this advice. We have decided to move this discussion of the comparison of methods to the beginning of the discussion (was section 4.3, now section 4.1).

*Lines 937-958: This paragraph could be pared down or removed. It is nice to understand the thinking behind omitting other carbonate system parameters but since they are not mentioned previously, it almost seems extraneous.*

**Response:**
We believe that it is a useful contribution to note the need for sensor data that can better calculate the full carbonate system. We decided to move this to the supplemental materials, lines 138 – 142.

*Conclusion: I recommend that the focus should be shifted to the significance of this study, rather than a summary of parameter-by-parameter results. The paragraph in lines 728-738 would fit in nicely. Lines 982-988 are strong conclusions that show the significance of the study as well.*

**Response:**
The ideas that were previously in lines 728-738 and 982-988 have been moved to the conclusion, and the conclusion has been restructured to only include the most significant findings.

---

## Referee Report (RR1)

**Intro:**

I will defer to the authors choice on this, but the intro as is may benefit from some brief additional explanation or clarification about the intertwined nature of the two key aspects of the study, 1- the carbonate chemistry variability, and 2- the air-sea fluxes of the estuary. As written, the air-sea fluxes feel like a somewhat of an afterthought. It might benefit authors to add some point to reiterate that they are interested characterizing carbonate chemistry to for two related reasons. To me, these seem like 1) to better understand the variation in seawater pH to provide context for ocean/estuarine acidification and its impacts and 2) to better understand the variation in seawater pCO2 to improve estimates of air-sea fluxes (both of which are under characterized, particularly in their region). This slight distinction might set readers up well for the rest of the paper, but nonetheless, I believe it could be sufficient as is as well.

**Methods:**

I appreciate the changes made to this section; it looks great.

Lines 132-134 the authors write "sampling was conducted every two weeks during summer and monthly during winter", but it is unclear to me at what frequency. The supplemental material notes that for sensor calibration purposes, this was done via replicate samples by these sensors, but some clarity on how this was done for the air-sea flux calculations would be nice in the main paper. E.g., "These discrete samples were collected by taking 1 discrete surface sample within the site every two weeks". These details seem pertinent to have in text because you go on to calculate air-sea fluxes from these samples, so knowing the exact details is important to inform interpretation of the spatial and temporal resolution of these fluxes.

Lines 229-230, 235 (the equations): change font to be consistent with font in the paper text and with equation 1.

**Results:**

This section has been greatly improved since the last version and is far more streamlined with the material moved to the supplement. There are a number of smaller improvements that could be made described below.

Figure 4 – only one of these plots has an x-axis labeled "season". Formatting issues with panel letter labels (c hard to find in the y-axis label). I might move the panel labels onto the figure.

Figure 5- the x-axis label of "DATE" should be removed, as it is redundant. The y-axis labels also could be cleaned up removing the (Day-Night) label since this is redundant on each figure as well. I might suggest simply adding this to the figure caption but will defer to author/editor preference. It might also be nice to add the year (Jan-Jun) that this was conducted in the figure caption for clarity.

Figure 6 – similarly, adding the year to the top panel in the figure description might help for clarity to remind readers where this timeseries fits within the bottom two panels

**Discussion**

In general, a great discussion, although very long (almost 10 pages). I would recommend trimming where possible but agree with the authors' overarching thoughts. I would also recommend sub-sub sections to break up the text if possible, given its overall length. For example:

*4. Discussion*

      *4.2 Factors controlling temporal variation in carbonate system parameters*

            *4.2.1 Temperature*

            *4.2.2 Biological controls*

            *4.2.3 Tidal controls*

            *4.2.4 Salinity*

            *Etc*

            *Etc*

This would be very helpful I think to keep readers from getting lost in the block of text, and allow those interested in particular aspects of the work to more quickly zero in on sections of interest.

A few additional specific recommendations below:

Lines 516-523: See my point above about methods. I feel that it is important for readers to know exactly how many discrete surface samples these air-sea fluxes were calculated from if you later make the point that there was no difference between seasonal air-sea flux estimates from single biweekly samples, and continuous sensors (an interesting finding).

Line 531: add references of studies citing challenges with k parameterization

Lines 535-540: Yes, I agree these are some of the major sources of the challenge in determining air-sea fluxes, and I appreciate the inclusion of these additional datapoints on what fluxes might be if data were treated differently.

---

## Author Response (AR2)

We thank referee Melissa Ward and the anonymous referee for their feedback throughout the review process that has resulted in a substantially improved manuscript.

**Referee #1: Melissa Ward, maward@ucdavis.edu**

**Intro:**

I will defer to the authors choice on this, but the intro as is may benefit from some brief additional explanation or clarification about the intertwined nature of the two key aspects of the study, 1- the carbonate chemistry variability, and 2- the air-sea fluxes of the estuary. As written, the air-sea fluxes feel like a somewhat of an afterthought. It might benefit authors to add some point to reiterate that they are interested characterizing carbonate chemistry to for two related reasons. To me, these seem like 1) to better understand the variation in seawater pH to provide context for ocean/estuarine acidification and its impacts and 2) to better understand the variation in seawater pCO2 to improve estimates of air-sea fluxes (both of which are under characterized, particularly in their region). This slight distinction might set readers up well for the rest of the paper, but nonetheless, I believe it could be sufficient as is as well.

**Response**:**

We added one sentence at the end of the introduction to make this intent clearer: "The characterization of carbonate chemistry and consideration of regional drivers can provide context to acidification and its impacts and improved estimates of air-sea CO2 fluxes".

**Methods:**

**I appreciate the changes made to this section; it looks great.**

Lines 132-134 the authors write "sampling was conducted every two weeks during summer and monthly during winter", but it is unclear to me at what frequency. The supplemental material notes that for sensor calibration purposes, this was done via replicate samples by these sensors, but some clarity on how this was done for the air-sea flux calculations would be nice in the main paper. E.g., "These discrete samples were collected by taking 1 discrete surface sample within the site every two weeks". These details seem pertinent to have in text because you go on to calculate air-sea fluxes from these samples, so knowing the exact details is important to inform interpretation of the spatial and temporal resolution of these fluxes.

**Response**: Sampling frequency has been clarified. It now reads "A single, discrete, surface water sample was collected every two weeks during the summer months and monthly during the winter months from a small vessel at a station near (<20 m from) the sensor deployment."

*Lines* 229-230, 235 (*the equations*): *change font to be consistent with font in the paper text and with equation* 1.

**Response**: The equation font has been changed to match the text.

**Results:**

This section has been greatly improved since the last version and is far more streamlined with the material moved to the supplement. There are a number of smaller improvements that could be made described below.

*Figure 4 – only one of these plots has an x-axis labeled "season". Formatting issues with panel letter labels (c hard to find in the y-axis label). I might move the panel labels onto the figure.*

**Response**: The misplaced "season" label has been removed, and figure layout has been changed to make sure that all plot labels are easily visible.

Figure 5- the x-axis label of "DATE" should be removed, as it is redundant. The y-axis labels also could be cleaned up removing the (Day-Night) label since this is redundant on each figure as well. I might suggest simply adding this to the figure caption but will defer to author/editor preference. It might also be nice to add the year (Jan-Jun) that this was conducted in the figure caption for clarity.

**Response**: The y-axes have been simplified (the daytime versus nighttime difference is now only described in the caption). The figure layout has been changed to make sure that all plot labels are easily visible. The figure caption has been altered to include the date range for these continuous data.

Figure 6 – similarly, adding the year to the top panel in the figure description might help for clarity to remind readers where this timeseries fits within the bottom two panels

**Response**: The figure caption has been altered to clarify the range of time for continuous versus discrete data. Plot labels have also been added to be more consistent with other figures.

**Discussion**

In general, a great discussion, although very long (almost 10 pages). I would recommend trimming where possible but agree with the authors' overarching thoughts. I would also recommend sub-sub sections to break up the text if possible, given its overall length. For example:

4. Discussion
4.2 Factors controlling temporal variation in carbonate system parameters
4.2.1 Temperature
4.2.2 Biological controls
4.2.3 Tidal controls
4.2.4 Salinity
Etc
Etc
This would be very helpful I think to keep readers from getting lost in the block of text, and allow those interested in particular aspects of the work to more quickly zero in on sections of

interest.

**Response**: We broke up section 4.2 into four subsections, similar to these suggestions.

**A few additional specific recommendations below:**

Lines 516-523: See my point above about methods. I feel that it is important for readers to know exactly how many discrete surface samples these air-sea fluxes were calculated from if you later make the point that there was no difference between seasonal air-sea flux estimates from single biweekly samples, and continuous sensors (an interesting finding).

**Response**: We edited the text to clarify sampling frequency. It now reads, "However, we found no significant difference (within any season) between  $CO_2$  flux values calculated with hourly sensor data versus single, discrete samples collected monthly to twice monthly (Table S2, Fig. 3)."

*Line 531: add references of studies citing challenges with k parameterization*

**Response**:**

The following two references have been added:

- Borges, A.V., and G. Abril. 2011. 5.04 Carbon Dioxide and Methane Dynamics in Estuaries. In *Treatise on Estuarine and Coastal Science*, ed. E. Wolanski and D. McLusky, 119-161. Waltham: Academic Press.
- Van Dam, B.R., J.B. Edson, and C. Tobias. 2019. Parameterizing Air-Water Gas Exchange in the Shallow, Microtidal New River Estuary. *Journal of Geophysical Research: Biogeosciences* 124: 2351-2363.

Lines 535-540: Yes, I agree these are some of the major sources of the challenge in determining air-sea fluxes, and I appreciate the inclusion of these additional datapoints on what fluxes might be if data were treated differently.

**Anonymous Referee #2**

Thank you for your careful consideration of and thoughtful responses to the reviewer comments. I think this revised manuscript is in much clearer, and the re-organization was helpful in terms of readability. I have a few additional minor comments, listed below.

*Line* 87 *and* 96: *Define the acronym ASC at the first mention of Aransas Ship Channel (line* 87, *instead of line* 96)

Response: Thank you for catching this. It has been fixed.

Line 328: Missing (before "Table S1)".

Response: Fixed.

**Line 349: The spring continuous data that differs in sign from the discrete data has a tremendous SD. Were there outliers in the data that could be removed to improve this stat?**

**Response**: This opposite sign was not over the same time period; it is noted in the text that the negative spring flux from discrete data is for the entire 5+ years of monitoring (while continuous data only spanned 10 months). The sign difference between methods that is reported is likely indicating that the continuous period was not representative of the longer 5+ years of monitoring. The spring flux from discrete data during only the continuous monitoring period had the same sign as the continuous data (Table S1), and the two were not significantly different (Table S2).

While there is a large standard deviation, that is also the case for the continuous data during other seasons (Table S1), and there do not seem to be any visually egregious outliers in the continuous spring fluxes (Figure 3A). Removal of  $pCO_2$  outliers was already conducted prior to flux calculations (Figures S2 and S3). Additional differences between methods could likely be due to sampling bias of the discrete data (continuous data captures more variability in  $pCO_2$ ) and the different ways in which windspeed data were aggregated (the use of mean hourly wind speeds would allow for much more drastic flux values in the continuous data than the daily mean wind speed used for calculation with discrete sample data), which would naturally result in higher SD for fluxes calculated from continuous data. Individually pulling outliers would not likely improve this comparison.

**Line 513: Which studies specifically? Would like to see citations here.**

**Response**:**

The following two references have been added:

Crosswell JR, Anderson IC, Stanhope JW, et al. Carbon budget of a shallow, lagoonal estuary: Transformations and source-sink dynamics along the river-estuary-ocean continuum. *Limnol Oceanogr.* 2017;62:S29-S45. doi:10.1002/lno.10631 Liu H, Zhang Q, Katul GG, Cole JJ, Chapin FS, MacIntyre S. Large CO2 effluxes at night and during synoptic weather events significantly contribute to CO2 emissions from a reservoir. *Environ Res Lett.* 2016;11(6):1-8. doi:10.1088/1748-9326/11/6/064001

Line 608: Specify which time scale – I believe you're talking about seasonal here. Saying "on seasonal time scales" instead of "on certain time scales" would make the contrast with diel time scales (the following sentence) clearer.

**Response**: This has been fixed.

Line 712: I think it's important to emphasize again that it supports the validity of long-term discrete sample collection, because short-term sampling would not capture interannual variability in the region.

**Response**: We have added a sentence to emphasize this: "Discrete data captured interannual variability, which could not be captured by the shorter-term continuous sensor data."

Supplemental line 68: This mention of "buildup of respirational products" had me wondering about biofouling in the cooler – was there any evidence of biofouling, either in the cooler or on the submerged sensors, during the biweekly maintenance? Important to consider how this would impact measurements over time, if so.

**Response**: There was no evidence of biofouling within the cooler or on the Sami-CO2 or SeaFET sensors within the cooler. This is the advantage of pumping the water from the surface of the ship channel into the cooler for measurements – the organisms did not survive that pumping process, but the water chemistry remained representative of the in situ surface waters. The YSIs that were deployed directly in the ship channel did experience some biofouling. We swapped them out every two weeks to make sure that biofouling didn't get bad enough to influence the measurements.

**Supplemental line 93: I believe this should be Table S6 instead of Table S5.**

**Response**: Thank you for catching this. It has been fixed.

**Supplemental lines 104-105: This is repetitive from the previous paragraph – potentially an editing error. Could include the discussion of offsets and SDs in the previous paragraph instead.**

**Response**: This was an editing error. We have condensed text from these two paragraphs.

Something to consider: there is a lot of overlap between the results/discussion of the continuous and discrete samples, and including both types of data is a bit clunky at times. Because you make the point that long-term discrete sampling schemes capture interannual variability and are representative of the continuous sampling data, you could just focus on the discrete sampling set in your results/discussion where possible (seasonal variability and controlling factors). Diel variability results/discussion would focus on the continuous data, of course. This might improve readability and remove excess length. There are some slight differences between continuous and discrete data, but since you conclude that discrete is representative of continuous, it would be okay to ignore those differences. This would necessitate a major revision, so it's just a suggestion.

**Response**: We appreciate this thoughtful suggestion. Due to the extensive nature of the edit and the exclusion of the few noted differences between methods, we decided to keep the continuous data in the results/discussion.

---

## Author Response (AR3)

**Responses to comments by the Associate Editor:**

*Line 72: What do you mean by 'varying conditions'?*

This was meant to indicate that sensors are not typically tested in environments with highly variable salinities and as a result, environments with highly variable salinity have been found to have poorer correspondence between sensor and discrete measurements than areas with more constant salinity (Sastri et al. 2019). We changed the text to say "highly variable salinities".

*Line 84: What do you mean by this?*

By 'known acidification' we meant evidence of acidification (based on previous sentences). We changed the text to say 'evidence of acidification'.

*Line 127: Did you calibrate the sensor to lab measured pH, or just apply an offset based on this value to the entire pre-calibrated dataset? I think it is best to calibrate the entire dataset to the sample measured in lab based off the code in Bresnahan et al.*

After reading this comment and re-reading Bresnahan et al., we realized that we misinterpreted their correction and actually went with a more simplistic correction based on lab measured "benchmark" pH samples that were analyzed spectrophotometrically (with purified mCP) and converted to in situ temperature using $CO_2SYS$ (using measured pH and DIC). This type of correction has been done before in Shadwick et al. (2019). We changed the text (and supplementary materials) to note this and added clarification that biofouling—which would be a likely driver of sensor drift and more necessitate the more advanced correction—was not experienced in this case. The text now reads:

"The average difference between sensor pH and discrete quality assurance samples measured spectrophotometrically in the lab was used to establish a correction factor (-0.05) across the entire sensor pH dataset. Note, this correction scheme was not ideal (Bresnahan et al., 2014) although less rigorous correction based on sensor and discrete pH values has also been used (Shadwick et al. 2019). Nevertheless, the overall good agreement between discrete and corresponding sensor pH values during the deployment period suggested that the SeaFET sensor remained stable. It is also worth noting that our monitoring setup remained free from biofouling during the 10-month period. We attribute this to the deployment design in which the high frequency movement of the pumping mechanisms in the diaphragm pump must have eliminated the influence of animal larvae."

*Reference:*
*Shadwick, E.H., M.A.M. Friedrichs, R.G. Najjar, O.A. De Meo, J.R. Friedman, F. Da, and W.G. Reay. 2019. High-Frequency $CO_2$ System Variability Over the Winter-to-Spring Transition in a Coastal Plain Estuary. Journal of Geophysical Research: Oceans 124: 7626-7642.*

*Lines 679-681: What about the influence of the freshwater carbonate chemistry? If I remember correctly the rivers have high TA concentrations which could help buffer changes in pH/pCO2.*

You are correct that the region has high river alkalinity (see lines 77-78 for that background). This ship channel area is far removed from the location of freshwater inflow (see Fig 1), but the river chemistry does lead to generally high buffer capacity throughout the entire estuary. We mention that the relatively small fluctuation seen in pH is likely (at least partly) due to the high alkalinity in the region (see lines 575-579).

We agree that it is worth mentioning the unique river alkalinity situation at this location in the text, especially since the subsections were restructured based on reviewer comments and this section is specific to salinity and freshwater inflow. The rivers still have lower pH and higher $p$CO$_2$ than the seawater endmember in this region despite their high TA, but pH and $p$CO$_2$ may not be as extreme in the rivers here compared to others. We have added the following text to the section:

"Though the river water still has elevated $p$CO$_2$ and depressed pH compared to the seawater endmember, the high riverine alkalinity (often higher than the seawater endmember) in the region results in relatively well-buffered estuarine conditions in MAE (Yao and Hu, 2017)."